# How disconfirmatory evidence shapes confidence in decision-making
Annika Boldt [1] ✉, Yishu Sun[1] & Kobe Desender [2]

When assessing our decisions, the normative strategy involves giving equal weight to each evidence sample when computing confidence. However, recent findings suggest that the brain tends to overweight decision-congruent information when forming confidence judgements (i.e., positive-evidence bias; PEB). Here, we re-analyzed nine datasets (total $N = 176$) from human participants who judged the average color of eight shapes and gave their confidence. This task precisely allowed us to disentangle the impact of choice-confirming and choice-conflicting evidence on the formation of confidence. Strikingly, participants overly relied on evidence that conflicts with their choice, contrary to the normative model and the PEB. To explain this response-incongruent evidence effect in the computation of confidence, we fitted an extended log-posterior-ratio for confidence model to our data and show that the same robust averaging principle that influences decisions also accounts for these confidence effects: incongruent evidence receives a stronger weight in the computation of confidence because it lies closer to the category boundary around which there is heightened sensitivity. In a preregistered experiment ($N = 32$), we then empirically demonstrate that an experimentally induced shift in the category boundary affects the computation of confidence in otherwise identical stimuli. We conclude that confidence depends on dis-confirmatory evidence due to downstream consequences from decision-making mechanisms.

Humans possess the remarkable ability to express confidence in the correctness of their own decisions and other internal cognitive processes—a phenomenon called *metacognition*. For instance, we might feel high confidence in recognizing the person across the street as an old friend, or low confidence when taking a left turn on our way navigating an unknown city. Such confidence signals can help optimize behavior and support learning especially in the absence of external feedback[1–3]. The higher confidence correlates with objective accuracy, the more valid this proxy for feedback becomes. However, confidence is of course no 'crystal ball'—if we were able to reliably detect all our mistakes, human behavior would be flawless. Indeed, studies have shown that confidence and accuracy correlate moderately at best[4,5]. Such findings sparked the interest into investigating how confidence is formed internally and which signals it is based upon. One key question in this line of research is whether choices and confidence are transformations of the same underlying variables as some normative models would predict (cf. signal-detection theory or SDT frameworks[6,7]), or whether instead confidence can be dissociated from the evidence governing choices[8,9]. Several recent studies provide support for the latter case: confidence being based on a suboptimal combination of evidence. More specifically, studies have found confidence judgments to be based predominantly on response-congruent information[10–21]. For example, in a degraded face/house task Peters and colleagues[17] used intracranial recordings to demonstrate that choices could be reliably predicted by neural activity associated with both the selected and unselected choice option, whereas confidence was only associated with the degree of neural evidence for the selected but not the unselected choice option.

Confidence's blindness to decision-incongruent evidence is often referred to as the *positive evidence bias*, after the experimental method used to trigger it: researchers selectively manipulated the evidence supporting the two available response options with the result that evidence supporting the chosen option (positive evidence) tends to be overweighted for confidence[11,13,21]. Here, we will use the more neutral term *response congruent evidence* (RCE effect[22]) to refer to the empirical observation that confidence depends more strongly on response congruent evidence. Ignoring response incongruent evidence can inflate confidence, thus leading to overconfidence which could then trickle down to cause more cognitive distortions due to participants adjusting behavior based on an inflated sense of confidence. Crucially, in this case such overconfidence would not be due to a bias parameter as some other models might implement it, but is inherent in the evidence processing. For a real-world example, take the study by Ortoleva and Snowberg, who found that overconfidence and extreme political ideologies were linked[23]. Relatedly, Rollwage et al.[24] showed that participants

[1]Institute of Cognitive Neuroscience, UCL, London, UK. [2]Brain and Cognition, KU Leuven, Leuven, Belgium. ✉e-mail: a.boldt@ucl.ac.uk

at both political extremes had impaired metacognitive monitoring. The authors link this reduced insight to a reduction in error correction, which could explain the uphold of radical beliefs. Given these implications for metacognitive control, it is important to understand the mechanisms behind the RCE effect.

In the present paper, we investigated the construction of confidence using a perceptual decision-making task that allowed high control over the evidence samples presented to participants[25]. Our reasoning behind this was that most prior studies used stimulus material where the evidence fell into two mutually exclusive categories (e.g., left vs. right movement; house vs. face images; two-dimensional signal detection). While having many

advantages such as being able to selectively manipulate positive evidence, these tasks are of limited use if we are trying to capture how participants sample evidence from a continuous stimulus space as is arguably often the case in real-world decision making. In the current work, we re-analyzed nine datasets, all using the same color-discrimination task in which evidence is continuously distributed around a category boundary and noisy stimuli need to be categorized by participants as either red or blue (Fig. 1A). With occasional auditory feedback, we ensured that participants adopt this category boundary as their decision criterion as best as possible. This task was originally developed by de Gardelle and Summerfield to study robust averaging in decision making[25]. The authors found that evidence

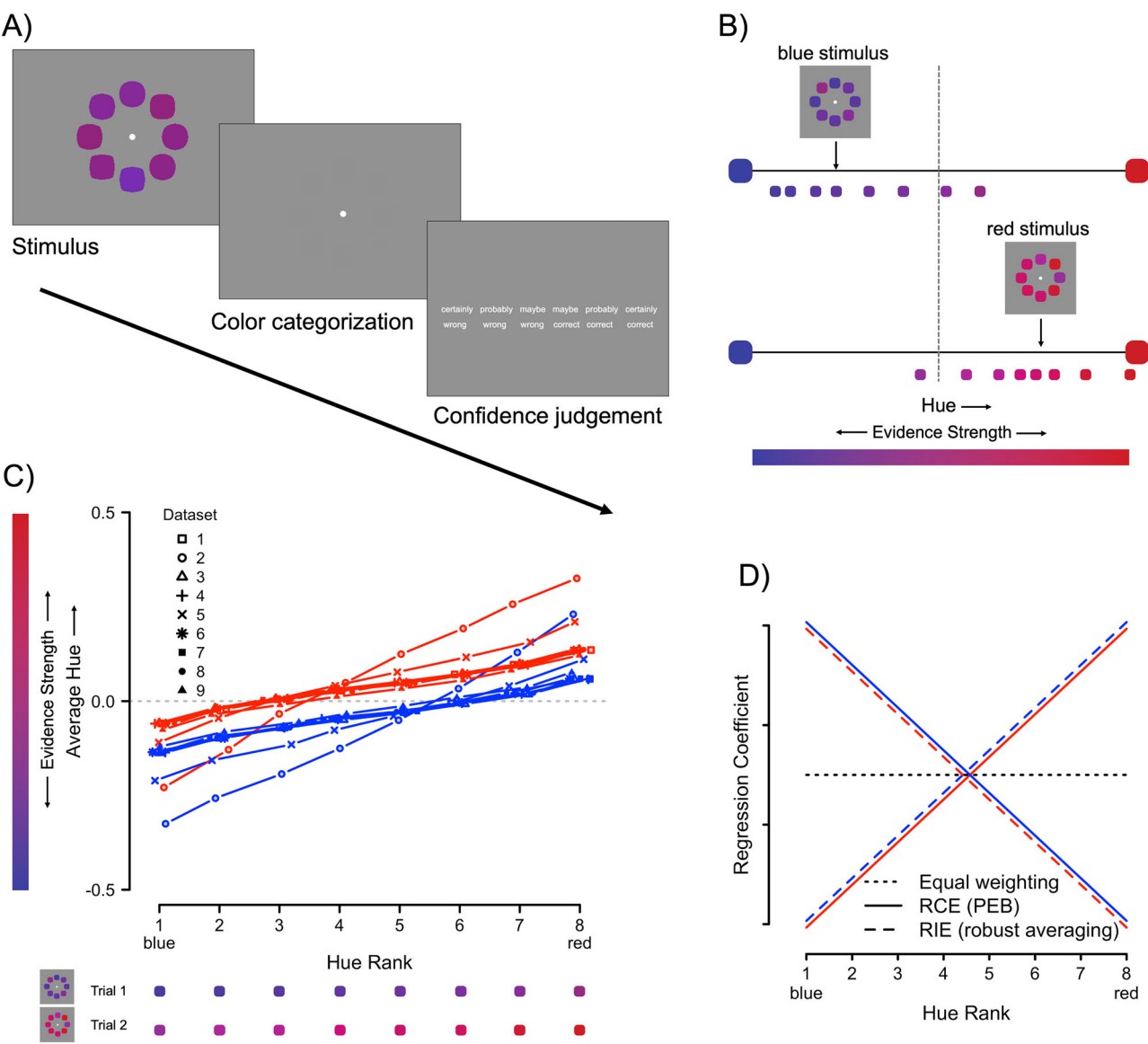

**Fig. 1 | Experimental design and stimulus characteristics. A** On each trial, participants indicated whether the average color of the eight elements was red or blue. After each choice, they rated their level of confidence on a six-point scale running from 1 ("certainly wrong") to 6 ("certainly correct"). **B** A blue (top) and red (bottom) example stimulus with its eight elements sorted according to hue. Note, arrows indicate the average color, the vertical dotted line reflects the category boundary. The strength of the (hue) evidence samples depends on the distance to the category (decision) boundary. **C** Characteristics of the stimulus sets: Average hue for the 8 positions (hue rank) across all 9 datasets as a function of stimulus category (blue or red, indicated by line color and dataset indicated by symbol). The horizontal dashed line reflects the category boundary at 0. Most stimulus sets intersected with the

category boundary at the third element (red stimuli) or the sixth element (blue stimuli). The example stimuli from (**B**) are shown below the x-axis with the elements now ordered from blue to red, to illustrate how the individual elements are used in the subsequent regression analyses. Uncertainty is reflected in the hairline error bars ($+/-1$ standard error of the mean), which are small and hence largely covered by the data points, average $N = 4562$ trials (min = 1141 trials; max = 10,073 trials).
**D** Possible results of a regression analyses predicting confidence based on the eight elements (sorted by color from blue to red; x-axis) for the three possible scenarios. RCE response-congruent evidence, RIE response-incongruent evidence, PEB positive-evidence bias. The predictions for blue and red trials (reflected in line colors) are slightly offset on the y-axis to avoid overlapping.

contributed to the decision-making process depending on its distance to the decision criterion: evidence samples falling closer to the decision criterion received a stronger weight in the decision (see also Vandormael et al.[26]; Ni and Stocker[27]). Robust averaging benefits decisions as it protects them against noise (Li et al.[28]; but see Van den Berg and Ma[29]).

Under standard SDT assumptions, it is expected that all elements contribute equally to the computation of confidence. This assumption is shown as the dotted, horizontal line in Fig. 1D. If on the other hand, our data were to reveal an RCE effect, this would be reflected in a pattern shown in Fig. 1D as the solid line: Under this account, we would expect confidence to mostly depend on the amount of red (vs. blue) in the stimulus when the category was red (vs. blue). As a third option, our data could reveal the inverse effect (dashed lines in Fig. 1D). This pattern follows directly from the robust averaging principle: In a stimulus where colored samples are distributed around a red mean, the blueish evidence lies, by definition, closest to the category boundary and vice versa (Fig. 1B). Figure 1C shows the average hues for each of the eight positions of the sorted elements (hue rank) as a function of color and dataset. In other words, for each stimulus we sorted the hue values of the eight composing elements to form their hue rank and then averaged across these eight ranks to visualize the distribution of the different elements. The category boundary (horizontal dashed line) intersects with the hue distributions roughly at the third rank position for red stimuli and the sixth rank position for blue stimuli. Since evidence close to the decision criterion (which should be close to the category boundary) is expected to contribute more strongly, the robust averaging principle therefore predicts that confidence excessively depends on variation in response-incongruent evidence rather than response-congruent evidence. Figure 1D shows a visualization of the three possible results. The strength of our task lies in its ability to precisely control the evidence samples upon which a choice is based. To foreshadow our results, across nine different datasets, we observed that confidence mostly depends on response incongruent evidence (i.e., supporting the RIE hypothesis). We then turn to adaptive gain theory, extending the computational model proposed by de Gardelle and Summerfield[25] to also include confidence judgments. Results from a formal model comparison show that our findings are best explained by a model which assumes heightened sensitivity for evidence lying close to the decision criterion. Such a model furthermore outperformed a competing model that implements the positive evidence bias (RCE effect). Importantly, the parameter controlling heightened sensitivity was responsible for explaining both robust averaging in choices as well as the RIE effect in confidence, thus providing a single explanation for both phenomena. Finally, we tested a prediction of our model in a preregistered experiment: Participants faced identical purple stimuli within two distinct color contexts: distinguishing between blue and purple (henceforth referred to as the 'Blue Context'), or purple and red ('Red Context'). We then go on to test whether the direction of the RIE effect could be reversed by the color context (i.e., for physically identical purple stimuli, confidence would be driven most by bluish elements in the Blue Context and reddish elements in the Red Context). We discuss how such a pattern could be reconciled with the RCE effect, indicating that they could stem naturally from different decision-making scenarios (one- vs. two-dimensional signal detection).

In summary, the present work tested three alternative hypotheses regarding how evidence contributes to confidence judgments in perceptual decision making. First, under standard signal detection theory assumptions, all evidence samples should contribute equally to confidence. Second, in line with the RCE effect, confidence might depend predominantly on variation of evidence supporting the chosen category. Third, consistent with the robust averaging principle, confidence could instead rely disproportionately on response-incongruent evidence lying closest to the category boundary,

leading to a data pattern in which confidence depends more on variation of the evidence supporting the unchosen category (RIE effect).

## Methods
### Paradigm and participants
In the present study, we re-analyzed 9 datasets that have been published previously[30–35] (total N = 176). All datasets had been collected using the same color-discrimination task originally described in de Gardelle and Summerfield[25] and extended for confidence use in Boldt and colleagues[30]. In this task, participants are instructed to quickly and accurately report whether the average color of an array of shapes is red or blue and then report their confidence that this decision was correct on a six-point scale running from 1 ("certainly wrong") to 6 ("certainly correct"; Fig. 1A). Task difficulty was manipulated in two orthogonal ways: by varying the distance of the mean color from the category boundary and by varying the variation across the color of the different elements. The colors of the eight shapes that composed each stimulus varied continuously from blue to red. Critically, individual elements within a stimulus could cross the category boundary. For instance, a stimulus that was on average blue could have individual elements that were red (Fig. 1B). In total, our pooled data included a total of 82,118 trials. Further details on the datasets can be found in Supplementary Note 1 and in Table S3.

We designed a follow-up experiment to test the model's prediction that closeness to the category bound is the deciding factor that causes the RIE effect. This experiment was preregistered on 28 April 2023 and the preregistration be found on https://osf.io/4326x/. We tested 37 participants, 5 of whom were excluded due to our preregistered criteria. All five had an error rate higher than 30%. No participants were excluded due to our other two exclusion criteria (more than 10% misses and selecting one of the confidence levels more than 90% of the time). Our final sample comprised 32 participants, in line with your preregistered sample size, based on the recommendation by Brysbaert and Stevens[36] to have at least 1600 observations per cell (resulting in a minimum of N = 11; 11*150 trials = 1650 observations). Twenty-seven participants were perceived by the experimenter to be women, and five to be men, based on apparent gender presentation, as gender identity was not formally recorded. Participants reported an average age of 22.8 years (19–30 years). Data on race or ethnicity was not collected. Participants reported having intact color vision and no psychiatric or neurological disorders. Our study lasted approximately one hour and participants were remunerated with £10. All testing was approved by the University College London ethics committee (1584/003) and all participants gave informed consent prior to taking part in the study.

We manipulated the category bound using a within-subject design: participants completed 5 experimental blocks in each color context, either deciding whether stimuli were blue versus purple (Blue Context) or purple versus red (Red Context). The purple stimuli used in both contexts were identical, but we expected to find that purple in the Blue Context would be treated like red (i.e., blueish elements contribute more to choices and confidence) and that purple in the Red Context would be treated like blue (i.e., reddish elements contribute more to choices and confidence). Stimulus categories were equally frequent within each context. Each decision was followed by a confidence judgment, again using a 6-point scale. Each experimental block was 84 trials long, 24 of which were trials without confidence judgments and instead auditory error feedback allowing participants to avoid developing a strong color bias. These feedback trials were excluded from all analyses. This means we included 300 trials per color context. The order of color contexts, as well as the color key mapping, and the confidence key mapping were counterbalanced.

We had planned to create red and blue stimuli as equidistant to the purple ones, therefore creating color contexts of comparable difficulty. However, due to a coding mistake, the red stimuli were closer to the purple than the blue stimuli, creating a more difficult Red Context. While unfortunate and likely to introduce more noise into our data, this mistake does not have any effect on our main contrasts given that the purple stimuli were

identical as planned. The main text reports only the preregistered key hypothesis. All other analyses can be found in Supplementary Note 2.

It should be noted that we deviated slightly from the preregistration: although we had originally planned to fit all data within a single model, concerns about multicollinearity led us to instead fit four separate models (one for each color and context condition) rather than using dummy variables. As a result, the slope comparisons for the purple trials in the two different contexts reported here are based on estimates from two separate linear mixed models, rather than on the interaction between context and hue that we had originally preregistered to test.

### Behavioral data analyses

Most analyses were performed using linear or logistic mixed models. All models included at least a random intercept per participant (nested within their respective experiment), and all manipulations of interest and their interactions as fixed effects. These models were then extended with random slopes in order of biggest increase in BIC, until the addition of random slopes led to a non-significant increase in likelihood or until the random effects structure was too complex to be supported by the data (leading to an unstable fit). We used the lmer and glmer functions of the lme4 package[37] to fit the linear and generalized linear mixed models, respectively, in R[38]. The calculation of $p$ values is based on chi-square estimations using the Wald test from the car package[39].

The following analyses were performed on the pooled data. First, we tested whether self-reported confidence shows a meaningful relationship with accuracy, by fitting a logistic mixed model in which trial-level accuracy was predicted by confidence. Second, we tested the influence of stimulus mean and variance on accuracy and confidence, by fitting a logistic and linear mixed models on these outcomes, respectively. Third, we tested for the previously reported finding[30,33] that when comparing conditions with high mean, high variance versus low mean, low variance, accuracy is matched whereas there is a clear difference in confidence. Note that we could test this specific finding only in seven datasets in which we used the specific $2 \times 2$ design. For these studies, we selectively analyzed confidence and accuracy in the low mean, low variance condition versus the high mean, high variance condition. To follow-up on the non-significant difference in accuracy, we computed a Bayes Factor with default priors as implemented in the BayesFactor package[40]. Fourth, to test for the robust averaging effect for choices[25], we fitted a logistic mixed model predicting choices ('red' vs. 'blue' response) with the hue values of the eight colored elements of each stimulus as predictors, sorted from red to blue, and for each element we also included its interaction with the stimulus category (i.e., red or blue) and for Dataset 10 also the color context. We then tested specific hypotheses on these models by constructing post-hoc contrasts, as implemented in the multcomp package[41], which tested the following hypothesis: (i) a first pair of contrasts tested, separately for red and blue stimuli, whether the beta coefficients showed a linear slope, (ii) a second pair of contrasts tested, separately for red and blue stimuli, whether inlying elements (positions 3–6) had stronger beta weights compared to outlying elements (position 1–2 and 7–8), and (iii) a third pair of contrasts tested, separately for red and blue stimuli, whether stimulus-incongruent elements (positions 1–4 for red, positions 5–8 for blue) had stronger beta coefficients than stimulus-congruent elements (positions 5–8 for red, positions 1–4 for blue). Fifth, we tested whether there was a similar robust averaging effect for confidence, by replicating the previous analysis using a linear mixed model with confidence as a dependent variable. This final behavioral analysis addressed our key question: whether our data would show a RCE effect or instead a RIE effect. The same contrasts as specified for the choice model were used to test specific hypotheses. However, the contribution of inlying versus outlying elements aimed to replicate a previously reported choice effect by de Gardelle and Summerfield[25] and was therefore omitted for confidence.

Both for the mixed models predicting choices and confidence, we chose to split the data by their objective stimulus category ('blue' vs. 'red'

trials), thereby matching the approach used by de Gardelle and Summerfield[25]. Importantly, our results do not change if instead we split by response (cf. Fig. S1). We therefore use the terms RCE and RIE throughout this paper rather than speaking of stimulus congruency.

Wherever confidence intervals for the fixed effects are shown, these are obtained directly from the mixed models. They reflect the estimated population-level effects, computed across participants while accounting for the nesting of participants within datasets through the model's random-effects structure.

For all $t$-tests, we formally assessed normality using Shapiro-Wilk tests. Unless explicitly noted otherwise, no violations of the normality assumption were detected. For all Bayesian paired-samples $t$-test we used default Cauchy priors ($r = 0.707$). For all Pearson correlations, we assessed linearity visually and tested for bivariate normality using Shapiro-Wilk tests. In cases where normality was violated, Spearman rank correlations were used instead and confidence intervals were estimated using a bootstrapping approach. Model convergence was verified for all reported linear and logistic mixed-effects models. Multicollinearity among fixed effects was assessed via variance inflation factors (VIF), with all models exhibiting acceptable values (VIF < 5). Full assumptions (e.g., normality of residuals, homoscedasticity) were assessed for a subset of models, but full diagnostics were not feasible for all due to model complexity and computational constraints. Given the large sample sizes, the models are expected to be robust to minor assumption violations.

### The c-LPR model

Our modeling efforts were based on the LPR model originally proposed by de Gardelle and Summerfield[25] (see also Yang and Shadlen[42]) for this specific multielement color averaging task. A detailed description of the model is given in supplementary information of de Gardelle and Summerfield[25]. The model assumes that the eight elements are passed through a sigmoidal transfer function:

$$E_i = -1 + \frac{2}{1 + e^{-\frac{C_i}{d}}} \tag{1}$$

where $C_i$ reflects the color (i.e., hue) value of element $i$, $d$ reflects the compression parameter and $E_i$ color element $i$ after the sigmoidal transfer. The individual $C$ elements are in hue color space and (theoretically) range between 0 and 1. However, for convenience, they have been rescaled between $-0.5$ and $0.5$ such that they are symmetrical around the category boundary. After the sigmoidal transfer, the $E$ values thus range between $-1$ and 1 (see Fig. 2C for a visualization of this transfer). Subsequently, the eight elements are averaged in a noisy fashion:

$$DV_1 = \frac{1}{8} * \sum_{i}^{8} E_i + N(0, \varepsilon_1) \tag{2}$$

with the noise assumed to come from a Gaussian distribution around zero and standard deviation $\varepsilon_1$ (i.e., reflecting decnoise), and $DV_1$ reflecting the decision variable used to compute a choice $x \in \{-1; 1\}$ by comparing its value to zero:

$$x = \{1, DV_1 > 0; -1, DV_1 < = 0\} \tag{3}$$

Going beyond the originally proposed LPR model, we assumed that this signal is then further corrupted by additional noise:

$$DV_2 = DV_1 + N(0, \varepsilon_2) * ((var_j - var_1) * \gamma) \tag{4}$$

With $\varepsilon_2$ reflecting metanoise, $\gamma$ reflecting scalemetanoise, $var_j$ reflecting the level of evidence variance in condition $j$ and $var_1$ the lowest level of evidence variance. Note that the influence of $\gamma$ itself depends on the difference in variance between condition $j$ and the

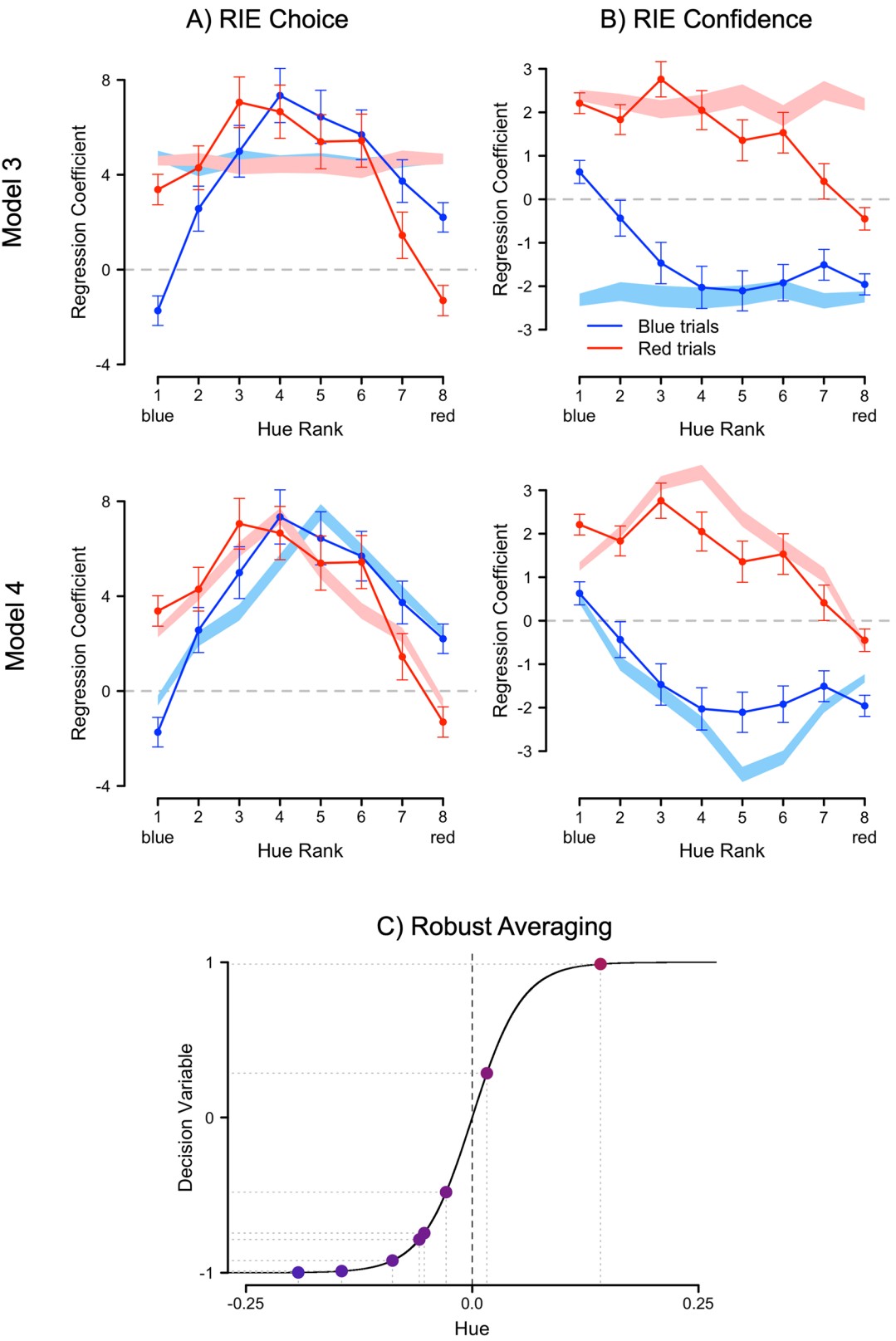

condition with the lowest amount of variance, that is, implying that no additional noise was added for the lowest level of variance, and the additional noise increased with increasing evidence variance. For the models without metanoise as a free parameter, we assumed that $\varepsilon_2 = \varepsilon_1$ (i.e., there was still metacognitive noise, but its amplitude was identical to that of decnoise).

With Model 7, we further chose to implement the RCE effect, or PEB. Specifically, we implemented a model where confidence is computed based on response-congruent evidence only. This model had the same two free parameters as Model 4 (i.e., the winning model) with the only difference being the way that confidence was quantified. Given that this model only considered choice-congruent evidence to compute confidence, it was

**Fig. 2 | Behavioral and simulated reliance (RIE) effects. A** Choice and **B** confidence effects as a function of stimulus category (red or blue). The top row depicts the results for Model 3, the bottom row for Model 4 (the winning model), each using the best-fitting parameter combinations for this respective candidate model. The stimulus-composing elements were ranked by their hue and the color for each of the resulting eight ranks used as a predictor for choice (**A**) or confidence (**B**), respectively. The resulting slopes of the regression coefficients indicate that both choices and confidence judgments were relatively more affected by response-incongruent evidence (RIE). In other words, for a red stimulus, the bluest elements showed the strongest effect and vice versa. The empirical data is depicted as lines with uncertainty reflected in the error bars (2.5% and 97.5% Wald confidence intervals; $N = 41,046$

observations for blue trials, $N = 41,072$ observations for red trials). The simulated data is depicted as bands with uncertainty reflected in the width of the bands (2.5% and 97.5% Wald confidence intervals; $N = 350,122$ observations). Whether the resulting regression weights are positive or negative depends entirely on how they were entered into the regression and is arbitrary. The focus lies on their relative contributions in the prediction. **C** The confidence extension of the log posterior ratio (c-LPR) model assumes robust averaging, meaning that instead of directly averaging the hues of which the stimulus is composed (in our experiment, hue values of the individual elements), these first get passed through a sigmoidal transfer function. Such a transfer leads to a compression of differences at the extremes (red and blue items) and augments differences at the category boundary (purple items).

necessary to add decision-noise to individual elements instead of to the group mean (as done for the other models). Thus, for the PEB model we first added noise to the individual elements:

$$E_i = C_i + N(0, \varepsilon_1) \quad (5)$$

Then, we transformed these values using the same compression function, $f$, as in the other models (i.e., Eq. 1) and then computed the mean:

$$DV_1 = \frac{1}{8} * \sum_{i=1}^{8} f(E) \quad (6)$$

For confidence, however, we computed the decision variable as

$$DV2 = \sum E_{congruent} + N(0, \varepsilon_2) \quad (7)$$

Where $E_{congruent}$ reflect choice-congruent elements.

Finally, confidence for all models was calculated on a six-point scale by comparing $DV_2$ to a fixed set of decision criteria. If $x$ equals to 1, the criteria were {Inf; 0.60; 0.35; 0.10; −0.15; −0.40; -Inf}. If $x$ equals to −1, the same criteria were used after multiplying these by −1.

When fitting the model to empirical data, we minimized the following cost function, separately for each participant, each level of evidence mean and each level of evidence variance:

$$SSE = \sum_{1}^{k} \sum_{1}^{j} \left( oACC_{j,k} - pACC_{j,k} \right)^2 + \sum_{1}^{k} \sum_{1}^{j} \left( oCJ_{j,k} - pCJ_{j,k} \right)^2 \quad (8)$$

With $oACC_{j,k}$ and $pACC_{j,k}$ reflecting, respectively, observed and predicted accuracy with evidence mean $k$ and evidence variance $j$, and $oCJ_{j,k}$ and $pCJ_{j,k}$ reflecting their counterpart for confidence judgments. Importantly, as can be seen in Eq. (5), we only optimized models to explain average accuracy and average confidence but did not directly fit the models to the RIE effect. Therefore, if the models are able to capture the RIE these effects naturally arise within that specific model, rather than the model being optimized to explain the specific effect. Models were fitted using a differential evolution algorithm[43], as implemented in the DEoptim R package[44]. The algorithm was run using 1000 iterations per model fit. AIC values for model comparison were computed as follows[45]:

$$AIC = 2k + n * \ln\left(\frac{SSE}{n}\right) \quad (9)$$

with $k$ the number of free parameters and $n$ the number of data points.

**Parameter recovery**. We simulated data from the winning model for one hundred agents for which we randomly selected a value for each of the two parameters. Subsequently, these data were fitted with the (winning) computational model, and correlations between true and fitted parameters were examined. We randomly selected values from a uniform distribution

for compression (range between 0.00001 and 0.2) and decnoise (range between 0.00001 and 0.3). As a sanity check, we first simulated a large number of trials (8000 trials per simulated agent), which provided excellent recovery for both compression, $r = 0.995$, $n = 100$, $p < 0.001$, 95% CI [0.992, 0.996], and decnoise, $r = 0.993$, $n = 100$, $p < 0.001$, 95% CI [0.989, 0.995]. All variables included in the Pearson correlations violated the normality assumption, as indicated by Shapiro-Wilk tests. Consequently, we repeated all analyses using Spearman rank correlations, resulting in similar results (compression, $\rho = 0.996$, $n = 100$, $p < 0.001$, 95% CI [0.993, 0.997], and decnoise, $\rho = 0.989$, $n = 100$, $p < 0.001$, 95% CI [0.984, 0.993]). We then repeated this process with only 250 trials per simulated agent, corresponding to the experiment with the lowest trial counts. Recovery for the two parameters remained excellent (compression: $r = 0.990$, $n = 100$, $p < 0.001$, 95% CI [0.985, 0.993]; decnoise: $r = 0.989$, $n = 100$, $p < 0.001$, 95% CI [0.984, 0.993]; see Fig. S2B). Once more we had to repeat the analyses using Spearman rank correlations, resulting in similar results (compression, $\rho = 0.991$, $n = 100$, $p < 0.001$, 95% CI [0.986, 0.994], and decnoise, $\rho = 0.987$, $n = 100$, $p < 0.001$, 95% CI [0.981, 0.992]).

**Model recovery analysis**. In addition to parameter recovery, we performed model recovery to test whether data generated under a specific model would also be best accounted for by the generative model. To do so, for each of the six models under consideration, we generated 50 datasets using the same parameter range as used during parameter recovery, and then fitted each of these with all six models. For each model fit, we computed the AIC value. The results from this parameter recovery were then summarized into Fig. S3F displaying the probability that the dataset was best fitted by (i.e., has the lowest AIC) the generative model compared to the other models.

**Mixed model analysis on simulated data**. We conducted analyses parallel to the ones described above in which we predict choices or confidence, respectively, from the color values of the ordered eight elements, split into red and blue colors. These logistic mixed models (choices) or linear mixed models (confidence) were estimated for the simulated data generated by the best-fitting version of each of the six models. We encountered singular fits for several of the choice models, namely Models 1, 2, 3, and 6. We assume the source of this problem to be low variability in the simulated data, so the mixed models fitted to the simulated choice results should be treated with caution in some cases.

**Reporting summary**
Further information on research design is available in the Nature Portfolio Reporting Summary linked to this article.

## Results
### Behavioral results

**Sanity checks**. Our first set of statistical analyses focused on the raw behavioral data. Before turning towards the key question, namely, how stimulus-congruent and stimulus-incongruent evidence drive confidence, we first aimed to demonstrate the overall high quality of the data. To do so, we demonstrated that in our data (i) confidence closely tracks

accuracy, (ii) accuracy and confidence both depend on the mean and variance of the stimuli, (iii) confidence differs for conditions matched in accuracy, and (iv) the robust averaging effect for choices originally reported by de Gardelle and Summerfield[25] can be replicated in our data.

As a first sanity check, we asked whether participants used the confidence scale in a meaningful way. To make the data from all studies directly comparable, we transformed the continuous confidence judgments from Desender and colleagues[33,35] to ratings from 1–6 or 4–6, respectively (see "Methods"). We then submitted single-trial accuracy to a logistic mixed model with confidence as a predictor and participants nested within their respective experiment. As expected, there was a significant fixed effect of confidence level, $\beta = -0.94$, $SE = 0.08$, $z = -11.96$, $p < 0.001$, 95% $CI$ $[-1.10, -0.79]$. This corresponds to an odds ratio of $OR = 0.39$, 95% CI $[0.33, 0.45]$: Mean accuracy was lowest for low confidence and highest for high confidence ($M_1 = 0.17$, $M_2 = 0.39$, $M_3 = 0.55$, $M_4 = 0.73$, $M_5 = 0.84$, $M_6 = 0.91$). We can therefore conclude that participants used the confidence scale in a meaningful way.

Second, we replicated the finding of previous work showing that both the mean and the variance of the stimulus influence performance[25,30,46]. To do so, we again ran a logistic mixed model predicting error rate, with participants nested under experiments. Both main effects of mean, $\chi^2(1) = 1816.94$, $p < 0.001$, and variance, $\chi^2(1) = 1103.67$, $p < 0.001$, were significant; but not their interaction, $\chi^2(1) = 1.23$, $p = 0.267$. The estimated coefficient for mean was $\beta = -0.57$ ($SE = 0.73$, $z = -41.99$, $p < 0.001$, 95% CI $[-32.00, -29.14]$), for variance, $\beta = 7.23$ ($SE = 0.23$, $z = 31.97$, $p < 0.001$, 95% CI $[6.78, 7.67]$), and for their interaction, $\beta = 1.25$ ($SE = 1.12$, $z = 1.11$, $p = 0.267$, 95% CI $[-0.95, 3.44]$). Because the predictors were continuous and varied over a narrow range, odds ratios based on a one-unit increase lacked interpretability and are not included. Error rates increased when stimulus strength was weak, and when stimuli were more variable. The lines and points in Fig. S3A (left column of panels) show a visual representation of these effects, averaged across studies when possible (i.e., when the same stimulus values were used). Confidence showed a similar pattern (shown in Fig. S3B) with both a main effect of stimulus mean, $\chi^2(1) = 1583.76$, $p < 0.001$, and stimulus variance, $\chi^2(1) = 32.02$, $p < 0.001$, and a significant interaction, $\chi^2(1) = 62.95$, $p < 0.001$. The corresponding coefficients were $\beta = 14.39$ ($SE = 0.48$, $t(73024.8) = 30.26$, $p < 0.001$, 95% CI $[13.46, 15.33]$) for mean, $\beta = -4.62$ ($SE = 1.05$, $t(8.6) = -4.42$, $p = 0.002$, 95% CI $[-6.67, -2.57]$) for variance, and $\beta = -29.36$ ($SE = 3.70$, $t(73043.1) = -7.93$, $p < 0.001$, 95% CI $[-36.62, -22.11]$) for their interaction. These findings reflected that confidence decreased when stimulus strength was weak, or stimuli were more variable, again replicating earlier studies[30,31,33,35,47].

Third, we replicated our original finding using this paradigm that when we carefully performance-matched two conditions of medium difficulty (low mean, low variance vs. high mean, high variance) confidence was reduced in the latter condition (for recent replications of this effect using different stimuli with a similar paradigm, see also Allen et al.[47]; Bang and Fleming[48]). Seven of our datasets had a similar 2 × 2 factorial structure. A logistic mixed model with participants nested under experiments predicting accuracy based on condition (low mean, low variance vs. high mean, high variance), confirmed that both conditions were not significantly different in accuracy, $\chi^2(1) = 3.69$, $p = 0.055$. The estimated coefficient was $\beta = 0.09$ ($SE = 0.05$, $z = 1.92$, $p = 0.055$, 95% CI $[-0.002, 0.19]$; $OR = 1.10$, 95% CI $[0.998, 1.21]$). To complement the trial-level analysis, we averaged accuracy per participant in each condition and submitted these to a Bayesian paired-samples $t$-test. This yielded a Bayes Factor of $BF_{10} = 0.35$, providing moderate evidence in favor of the null hypothesis (i.e., no accuracy difference between conditions). A mixed model on single-trial confidence ratings with condition as predictor did show a significant main effect of condition, $\chi^2(1) = 271.09$, $p < 0.001$. The estimated coefficient was $\beta = 0.18$ ($SE = 0.01$, $t(35470) = 16.46$, $p < 0.001$, 95% CI $[0.16, 0.20]$). Confidence was indeed lower in the high-mean high-variance condition ($M = 4.67$) compared to the low-mean low-variance condition ($M = 4.85$). It should be noted that this effect could not be explained by the small, non-significant difference in accuracy, which was in the opposite direction with numerically higher

accuracy in the high-mean high-variance condition ($M = 0.82$) compared to the low-mean low-variance condition ($M = 0.81$).

Fourth, we found the robust averaging effect for choices. This effect was previously reported by de Gardelle and Summerfield[25] who found that the influence of more extreme (clearly blue or clearly red) items was downweighed when making choices relative to less extreme (purple) items. The lines in Fig. 2A—here superimposed onto simulations generated from two out of seven models further described below—show the resulting parameters from two logistic mixed models fitted to all pooled data predicting choices based on the hue values of the eight colored elements of each stimulus, separately for each color category (i.e., red or blue). As a consequence, the regression weight for the first element captures the extent to which *variation* in the bluest element (i.e., how "blue" the bluest element was) is associated with variation in the dependent variable, in the case the choice. The focus of our analyses then lies on how much the regression weights for the different hue ranks differ from zero relative to each other. The regression weights of the sorted color elements were all significantly different from zero, for both blue, $|zs| > 5.33$, $ps < 0.001$, and red trials, $|zs| > 2.90$, $ps < 0.004$ (full statistical reporting can be found in Table S4). A planned contrast confirmed that inlying elements (positions 3–6) contributed more strongly to the choice then outlying elements (positions 1–2 and 7–8), both for red stimuli ($\beta = 26.72$, $SE = 1.42$, $z = 18.80$, $p < 0.001$, 95% CI $[23.34, 30.10]$) and for blue stimuli ($\beta = 30.99$, $SE = 1.41$, $z = 21.92$, $p < 0.001$, 95% CI $[27.63, 34.36]$) reflecting that the regression coefficients for the more extreme elements were significantly lower compared to the more purple elements. A second planned contrast showed an effect of congruency, with category-incongruent evidence driving choices more than congruent evidence, both for red stimuli ($\beta = 10.41$, $SE = 0.79$, $z = 13.17$, $p < 0.001$, 95% CI $[8.53, 12.29]$) and blue stimuli ($\beta = 4.90$, $SE = 0.79$, $z = 6.20$, $p < 0.001$, 95% CI $[3.02, 6.78]$). It should be noted that the original study[25] reported the inlying versus outlying effect averaged across color categories, which presumably masked the effect of congruency.

**Confidence depends on stimulus-incongruent evidence.** Having demonstrated that our data are of high quality, by replicating four key findings in the literature, we now turn to the main question under investigation. Specifically, as a key analysis, we next tested the extent to which stimulus-congruent and stimulus-incongruent evidence predicts confidence. The lines of Fig. 2B show the resulting regression weights from a mixed model predicting confidence, fitted to all pooled data taking into account the nesting of participants in experiments. Again, all sorted color element predictors were significantly different from zero, for both blue, $|ts| > 2.07$, $ps < 0.038$, and red trials, $|ts| > 1.98$, $ps < 0.047$ (full statistical reporting can be found in Table S5). Firstly, we note that most of the regression coefficients were positive. Such positive coefficients reflect that when an element from a red stimulus was very red, participants reported higher confidence compared to when this element was less red (and likewise for blue stimuli). However, the more striking pattern in our data was that the size of the regression coefficients markedly differed between the different elements. In Fig. 2B, the empirical regression weights are plotted as a function of color category and any effects are expressed in the slope which these regression weights form (cf. Fig. 1D). It should be stressed that by slope we do not refer to the individual regression weights, but instead to the patterns that are formed by the resulting lines of those weights. The slopes of the regression coefficients in Fig. 2B reveal clear qualitative differences between the different elements; the data show a clear negative slope for red elements and a clear positive slope for blue elements, demonstrating that stimulus-incongruent evidence has a very strong influence on the computation of decision confidence, whereas this is much less the case for stimulus-congruent evidence, showing a clear RIE effect. Thus, when participants saw a red stimulus, they reported a lower level of confidence when the bluest element was very blue compared to when it was purple, whereas the precise hue of the reddest element (i.e., very red or purple) did not have a strong influence on the level of confidence. The opposite logic holds for blue

stimuli: variation in the reddest elements drives confidence as opposed to the bluest elements. To formally back-up this claim, we constructed planned contrasts on the fit to test three specific predictions. First, a linear contrast confirmed the negative slope for red stimuli ($\beta = 30.10$, $SE = 0.62$, $z = 48.58$, $p < 0.001$, 95% CI [28.62, 31.57]) and the positive slope for blue stimuli ($\beta = 24.91$, $SE = 0.64$, $z = 39.20$, $p < 0.001$, 95% CI [23.40, 26.42]) of the beta coefficients. Second, a contrast comparing the weight of the incongruent elements versus congruent elements confirmed that incongruent elements had a much stronger weight in the computation of confidence, both for red stimuli ($\beta = 6.00$, $SE = 0.32$, $z = 18.65$, $p < 0.001$, 95% CI [5.24, 6.77]) and for blue stimuli ($\beta = 4.19$, $SE = 0.33$, $z = 12.55$, $p < 0.001$, 95% CI [3.40, 4.98]). In sum, our data clearly showed the inverse of the positive evidence bias, but was fully in line with the pattern predicted by the robust averaging principle: variation in the more stimulus-incongruent samples had a stronger effect on confidence compared to the stimulus-congruent samples (RIE effect).

All key analyses were performed collapsed across datasets, while taking into account the nesting of participants within experiments. Note, however, that the results were also highly consistent within each of the datasets. Indeed, when applying the contrasts to mixed effects models fitted separately to each dataset, we found strong evidence for significant slopes for both red and blue stimuli in all datasets, $ps < 0.007$ (Table S6), and we found strong evidence that incongruent elements contributed more than congruent elements for both red and blue stimuli in most datasets, $ps < 0.005$, except for blue stimuli in three datasets and in Dataset 7 there was no significant effect of either (see Table S7 for full details).

Jointly, these results suggest that *variation* in stimulus-incongruent evidence matters most for confidence. Intuitively, when a participant saw a blue stimulus, their confidence judgments will mostly reflect the amount of redness in the stimulus, whereas the amount of blueness in the stimulus does not really affect decision confidence. Although the previous analyses showed that *variation* in stimulus-incongruent evidence mostly drives confidence— as is common for all regression-based approaches—this does not show that confidence mostly depends on stimulus-incongruent elements themselves. We therefore used a stepwise regression approach in which we predicted confidence based on only seven colored elements, each time dropping one of the elements. The explained variance $R^2$ clearly decreased when stimulus-incongruent evidence was excluded whereas it remained almost the same when excluding stimulus-congruent elements (see Fig. S4). Together, these results overwhelmingly support the finding that confidence depends mostly on stimulus-incongruent evidence.

At this point, it is important to reiterate that in all previous analyses, the elements were sorted according to the color of stimulus (i.e., blue vs. red) and not according to the response provided by the participants. This choice was made because it allows for a straightforward comparison between the regression models for choice (as conducted by de Gardelle and Summerfield[25]) and regression models for confidence, given that these are based on the same data. Importantly, however, this analysis choice does not invalidate our claim that confidence mostly depends on response-incongruent evidence (RIE): the results presented in this section remain qualitatively similar when instead sorting the elements according to the response provided by the participant (see Fig. S1). For simplicity, we decided to use the terms stimulus congruency, response congruency, and decision congruency interchangeably throughout this study. We appreciate, however, that this is a simplification that might not apply to other studies.

**Modeling results.** To shed light on the underlying mechanism causing the RIE effect, we fitted several computational models to our data. The starting point of our modeling approach was the log posterior ratio (LPR) model originally proposed by de Gardelle and Summerfield[25]. The essence of this model is that, rather than directly computing a decision variable (DV) based on the average raw hue values of the colored elements of which the stimulus is composed, hue values are first passed through a sigmoidal transfer function and only then averaged to form the DV. The most important consequence of this sigmoidal transfer is that

elements far away from the category boundary are pushed closer to each other, while elements close the category boundary are artificially pulled apart. This non-linear transfer is visualized in Fig. 2C which plots the hue values of an example trial on the x-axis against the transformed values on the y-axis. As can be seen, there are two strongly blue elements, −0.19 and −0.14, which are at a sizable distance from each other in hue space, but after the sigmoidal transfer they are mapped onto −0.99 and −0.98 in DV space. As a consequence, variation in these elements is effectively abolished. Now compare this to two elements that lie close to the boundary with a hue value of −0.02 and 0.017. After the sigmoidal transfer these are mapped onto −0.48 and 0.29 in DV space, and thus their difference has been artificially increased. Given this example, it is intuitive to see why the sigmoidal transfer function effectively abolished variation in the extreme elements, whereas variation in elements close to the category boundary increased. The original LPR model has two parameters: Firstly, the slope of the sigmodal transfer function, which effectively controls the extent to which outlying samples are downweighted. Here we refer to this parameter as the "compression", to avoid confusion with the slopes in the data pattern formed by the regression weights for the different elements sorted by hue rank and reported above. Secondly, there is a decision noise parameter, which we will refer to as "decnoise". The LPR model successfully captures the downweighting in choices[25], however, this model has not yet been extended such that it can also capture decision confidence. Therefore, it remains unclear whether it can explain the reliance of confidence on stimulus-incongruent evidence. For this purpose, we designed a family of computational models that extend the original LPR model in various ways, in order for the model to also account for decision confidence.

**LPR confidence extension.** To model confidence within the LPR framework, we assumed that the mean decision variable (DV; which is computed after the sigmoidal transfer), does not just inform the choice (Fig. S3G) but also the reported level of confidence. To achieve this, we compared the DV to a fixed set of 5 confidence criteria, which mapped the DV onto a six-choice confidence scale (see Fig. S3H). Note that these confidence criteria were not free parameters, but instead were fixed across participants. Apart from the compression and decnoise parameters discussed earlier, we tested the importance of two additional parameters in explaining decision confidence and the RIE effect (see "Methods" for full details). First, we considered the necessity of an additional metacognitive noise parameter (termed "metanoise"), which adds an additional amount of noise to the DV *after* reading out the choice but *before* computing confidence. The choice of this parameter is inspired by several recent demonstrations showing that it is possible to dissociate noise at the choice level and noise at the metacognitive (i.e., confidence) level[49,50]. Second, we considered the necessity of a parameter which controls the extent to which metanoise depends on the level of evidence variance in the data (which we refer to as "scalemetanoise"). Previous work has demonstrated that stimulus variance has a stronger influence on confidence than it has on choices[30], which we sought to explain by assuming variance-dependent metacognitive noise. The rationale behind this parameter is therefore that metacognitive noise might be higher when there is more stimulus variance (i.e., scalemetanoise = 0 implies that metanoise is independent of stimulus variance).

**Competing models.** Table 1 lists the seven different competing models that we tested in the current work. Model 4 was the simplest model that we considered, which was essentially the LPR model with the addition of confidence as a further degraded variant of the decision variable *DV* (2 free parameters: compression and decnoise). Note that in this model the amount of metacognitive noise that was added to the signal before reading out confidence was assumed to be the same as the amount of decision noise (i.e., controlled by decnoise). Model 1 was the most complex model we considered, which features all four free parameters just described (4 free parameters: compression, decnoise, metanoise,

## Table 1 | Model comparison results

| Model | Parameter | | | | | Hardcoded RCE | df | Mean AIC | ∑ AIC |
| | Compression | Compression | Decnoise | Metanoise | Scalemetanoise | | | | |
|---|---|---|---|---|---|---|---|---|---|
| 1 | ✓ | – | ✓ | ✓ | ✓ | – | 4 | −35.23 | −6201 |
| 2 | ✓ | – | ✓ | ✓ | – | – | 3 | −36.65 | −6451 |
| 3 | – | ✓ | ✓ | ✓ | – | – | 3 | −33.58 | −5911 |
| **4** | **✓** | **–** | **✓** | ***** | **–** | **–** | **2** | **−37.30** | **−6564** |
| 5 | – | ✓ | ✓ | * | ✓ | – | 3 | −34.16 | −6011 |
| 6 | – | ✓ | ✓ | * | – | – | 2 | −34.48 | −6069 |
| 7 | ✓ | – | ✓ | * | – | ✓ | 2 | −28.11 | −4947 |

We compared seven different models in their ability to fit average accuracy and average confidence in each of the cells of the design. Model 4 is the winning model. Note that for Models 4–7, we did assume metacognitive noise to be present (indicated by *), but its value was fixed to be the same value as decnoise. Model 7 incorporates reliance of response-congruent evidence (RCE).
Bold formatting indicates the winning model.
*AIC* Akaike information criterion.

scalemetanoise). In Model 2, we dropped the assumption of scalemetanoise (i.e., setting its value to zero), but did estimate metanoise. In Model 3, Model 5, and Model 6, we implemented a linear (rather than a sigmoidal) compression parameter—implying that hues are linearly mapped onto the $[-1, 1]$ range before being averaged into the DV. Finally, in Model 7, we implemented the positive evidence bias, or RCE effect. To calculate confidence, each individual element is first distorted by noise. The model then discards all decision-incongruent elements and calculates confidence based on this *raw* response-congruent *stimulus* evidence (i.e., untransformed evidence). These changes only affect confidence, whereas choices continue to be based on compressed evidence. These models allow to directly test the importance of the compression parameter in explaining the RIE effect in choice and confidence.

**Quantitative model comparison**. In order to adjudicate between these different model variants, we fitted each model to the empirical data. Specifically, we tested each model's ability to explain average accuracy and average confidence in each of the cells of the experiment (see "Methods"). As a quantitative assessment of model fit, the right columns of Table 1 present average and summated Akaike information criterion (AIC) values across participants and experiments for each of these models. Model 4 (featuring compression and decnoise as free parameters) provides the best fit at a quantitative level, with a difference in average AIC of 0.64 and summed AIC of 113 compared to the second-best model (Model 2). When inspecting these quantitative results, it becomes clear that models without the compression parameter perform worse ($\Delta$AICs > 2.81 for average, and $\Delta$AICs > 495 for summed) compared to Model 4. As can be seen in Fig. S3A and S3B, most models do a decent job at capturing average accuracy and average confidence in each of the cells of the experiments, and they do so across experiments (Fig. S5) and individual participants (Fig. S6). See also Fig. S3C, which shows that the models captured the resulting proportions of error and correct trials for each confidence bin very well. A notable exception is Dataset 9, for which every model fails to provide an adequate fit for confidence. We assume this is due to the fact that in this study[33], a continuous confidence scale was used which, despite mapping it onto the scale used elsewhere (4–6), was not captured by our models very well. This interpretation is further confirmed by the finding that most models did capture average accuracy rather well for this dataset.

We have furthermore performed an analysis of differences between AIC indices. Specifically, we predicted AIC indices using a linear mixed effects model with a random slope for subjects, nested within experiments and with model as a fixed effect. Unsurprisingly, this model showed a main effect of model, $F(6, 1050) = 118.30$, $p < 0.001$, partial $\eta^2 = 0.40$, 95% CI

$[0.37, 1.00]$. More interestingly, we next constructed post-hoc contrasts comparing the winning model (Model 4) to the other models. This showed that Model 4 had significantly lower AIC values compared to models 1, 3, 5, 6, and 7, $ps < 0.001$ (see Table S8 for full statistics), whereas the fit indices were only numerically lower compared to Model 2, $p = 0.0995$. However, a Bayesian paired-samples $t$-test yielded a Bayes Factor of $BF_{10} = 1290$, providing extremely strong evidence in favor of Model 4. It should be noted that although inspection of the differences indicated deviations from normality, the sample size was sufficiently large ($N = 176$) that the Bayesian $t$-test is considered robust to such violations, and the results are therefore likely to remain valid.

**The compression parameter explains the RIE effect**. Although quantitative model comparison is very informative, it cannot tell us whether the models are able to capture key qualitative signatures in the data[51]. Specifically, given that the models were optimized to explain average accuracy and average confidence, the AIC values are uninformative with regard to the question whether the winning model can also explain the reliance on stimulus-incongruent evidence in the computation of confidence. To inspect this possibility, we simulated data from the different models and compared model predictions to the patterns seen in the empirical data (see Fig. 2A, B for Models 3 and 4, and Fig. S3D, E for all models). In particular, Fig. 2B shows that Model 4 closely captures the critical finding of the current manuscript; both in the data and in the model fit we see that confidence is mostly computed based on stimulus-incongruent evidence, whereas variation in stimulus-congruent evidence does not bear much weight (see also Table S9 for full statistics). Indeed, when applying the same contrasts to the results of the mixed model analysis previously performed on the empirical data to data simulated under Model 4, we observed positive slopes for blue ($\beta = 23.39$, $SE = 0.25$, $z = 94.24$, $p < 0.001$, 95% CI [22.80, 23.98]) and negative slopes for red stimuli ($\beta = 23.31$, $SE = 0.25$, $z = 93.76$, $p < 0.001$, 95% CI [22.72, 23.90]), stronger weights for incongruent versus congruent elements both for blue ($\beta = 5.45$, $SE = 0.13$, $z = 43.47$, $p < 0.001$, 95% CI [5.15, 5.75]) and red stimuli ($\beta = 5.37$, $SE = 0.13$, $z = 42.59$, $p < 0.001$, 95% CI [5.07, 5.67]), and stronger weights for inlying than for outlying elements both for blue ($\beta = 12.07$, $SE = 0.22$, $z = 55.29$, $p < 0.001$, 95% CI [11.55, 12.59]) and red stimuli ($\beta = 11.96$, $SE = 0.22$, $z = 54.83$, $p < 0.001$, 95% CI [11.44, 12.48]). At this point, we want to stress again that the fact that Model 4 reproduces the RIE effect is far from trivial as the models were merely optimized to fit average accuracy and average confidence; they were not fit to the regression data. Rather, the regression lines (i.e., the RIE effect) naturally fall out of Model 4. For completeness, the estimates of the free parameters in Model 4 can be found in Fig. S2A.

Inspection of the different models (cf. Fig. S3) reveals that the compression parameter is key in explaining the reliance effect. The three models that do not contain the compression parameter (i.e., Models 3, 5, and 6) not only provide a substantially lower quantitative fit, also qualitatively these three models completely miss the reliance effect in confidence; further demonstrating the importance of robust averaging in explaining the reliance effect in confidence.

**Variance-dependent metanoise explains confidence-accuracy dissociations.** As discussed earlier, confidence differs between two performance-matched conditions (i.e., high mean, high variance vs. low mean, low variance; the medium-condition effect). Importantly, this finding constitutes a separate effect independent of the occurrence of the RIE effect (reflected in the compression parameter) and is hence reflected in a different parameter in our model (scalemetanoise). Interestingly, although the model comparison favors Model 4 over the other models in terms of quantitative fit, it does seem to fall short in explaining the medium-condition effect. Indeed, the predictions from Model 4 show that the high mean, high variance and the low mean, low variance conditions were not significantly different in terms of accuracy, $\chi^2(1) = 0.13$, $p = 0.716$. The corresponding fixed-effect estimate was $\beta = 0.02$, SE $= 0.07$, $z = 0.36$, $p = 0.716$, 95% CI [-0.10, 0.15], which translates to an odds ratio of 1.02 (95% CI [0.90, 1.16]). To complement this trial-level result, we ran a Bayesian paired-samples $t$-test on subject-level average accuracies, which yielded a Bayes Factor of $BF_{10} = 0.40$, providing moderate evidence for the null hypothesis (i.e., no accuracy difference between conditions). Furthermore, the conditions were also not different from each other in terms of confidence, $\chi^2(1) = 0.40$, $p = 0.525$. The corresponding fixed-effect estimate was $\beta = 0.02$, SE $= 0.03$, $t(4.3) = 0.64$, $p = 0.558$, 95% CI [-0.03, 0.07]. A separate Bayesian paired-samples $t$-test resulted in a Bayes Factor of $BF_{10} = 0.17$, providing moderate evidence in favor of the null hypothesis (i.e., no confidence difference between conditions). It should be noted that for the second Bayesian paired-samples $t$-test, although the Shapiro-Wilk test indicated non-normality in the difference scores ($W = 0.97$, $p = 0.002$), this is unlikely to have impacted the results given the large sample size ($N = 149$) and the robustness of the Bayesian $t$-test to such deviations.

Further inspection of the different models reveals that only the models including scalemetanoise can explain this dissociation. Indeed, simulations from Model 1 mirrored the empirical finding that accuracy did not significantly differ between the two medium conditions, $\chi^2(1) = 0.60$, $p = 0.438$ ($\beta = 0.04$, SE $= 0.05$, $z = 0.78$, $p = 0.438$, 95% CI [−0.06, 0.14], OR $= 1.04$, 95% CI [0.94, 1.15]). A separate Bayesian paired-samples $t$-test supported this result, yielding a Bayes Factor of $BF_{10} = 0.34$, providing moderate evidence in favor of the null hypothesis. In contrast, the model predicted a significant difference in confidence between conditions, $\chi^2(1) = 8.05$, $p = 0.005$ ($\beta = 0.17$, SE $= 0.06$, $t(5.8) = 2.84$, $p = 0.031$, 95% CI [0.05, 0.29]). Here, the Bayes Factor was $BF_{10} = 601,926.3$, providing extreme evidence in favor of the alternative hypothesis. It should once more be noted that for the second Bayesian paired-samples $t$-test, although the Shapiro-Wilk test indicated non-normality in the difference scores ($W = 0.96$, $p < 0.001$), this is unlikely to have impacted the results as argued above. Thus, our modeling efforts explain the medium-condition effect as a selective influence of stimulus variance on metanoise *over and above* its effect on decision noise. Note, however, that this effect was rather subtle and therefore at the quantitative level Model 4 won the competition despite not being able to capture this effect. Indeed, if we restrict model selection to only Dataset 1[30] where conditions were performance-matched carefully to create the medium-conditions contrast, we find that a model including scalemetanoise is selected (i.e., Model 1).

**No flexible metanoise parameter.** A final takeaway of our modeling efforts is that our data favored a model assuming that metanoise takes the same value as decnoise, such as in the winning model (Model 4).

Models including a separate (i.e., flexible) parameter to control for metanoise (Models 1–3) perform worse. Several recent studies have investigated whether noise at the choice and confidence level could be dissociated[49,50]. Our findings speak against such an additional noise source.

**Empirically testing the model's prediction.** The c-LPR models with the compression parameter showed superior fit, letting us conclude that the closeness to the category boundary is the deciding factor underlying the RIE effect. Based on this observation, a crucial prediction from our model is that the RIE effect should entirely depend on the position of the category boundary. However, this is simply a prediction made by the model and therefore needs to be tested empirically. As described in the "Methods", we therefore designed a preregistered experiment in which participants categorized the exact same purple stimuli (consisting of eight colored elements) twice: once in the context of blue stimuli composed of eight elements (i.e., Blue Context) and once in the context of red stimuli composed of eight elements (i.e., Red Context; Fig. 3A). Contrasting the exact same purple elements dependent on the color context they appear in allows us to see whether physically identical elements will be assigned different confidence weights once the task-relevant feature space is shifted. In other words, do participants really base their confidence more on incongruent elements, regardless of where the elements fall in the color space? The key results of this experiment are presented in Fig. 3B: First, we replicated the RIE effect in this new dataset (cf. Supplementary Note 2). Second, as hypothesized, the direction of the RIE effect depended on the color context (Hypothesis 4). In the Blue Context, confidence for the purple stimuli over-relied on the 'blueish' items. In the Red Context, on the other hand, confidence for the purple stimuli over-relied on the 'reddish' items. Post-hoc contrasts indeed found a significant negative slope for purple trials in the Blue Context, $\beta = 45.11$, $SE = 11.54$, $z = 3.91$, $p < 0.001$, and a positive slope for purple trials in the Red Context, $\beta = 39.88$, $SE = 11.72$, $z = 3.40$, $p < 0.001$. The error bars in Fig. 3B are noticeably larger than those in Fig. 2. This is unsurprising, as the data in Fig. 3B is based on only $N = 32$ participants, whereas the previous analysis in Fig. 2 was conducted on a larger sample of $N = 176$ participants. The full statistics together with all our other preregistered analyses including results for the blue and red categories are listed in Supplementary Note 2 (see Table S3 and Figs. S7 and S8).

## Discussion

In the current work, we analyzed nine datasets in which participants categorized colors and rated their confidence in their decisions. In contrast to previous findings, which suggested that confidence mostly depends on response congruent evidence (RCE), our results showed that people tended to rely excessively on response incongruent evidence (RIE) when computing confidence in this task. Specifically, when a color stimulus had an average red hue, participants were more influenced by the bluest samples, rather than the reddest samples, when computing confidence. This effect was also present on the decision level, replicating a previous finding by de Gardelle and Summerfield[25]. Our findings can be explained by the robust averaging principle: the finding that evidence samples closer to the category boundary receive a stronger weight in the computation of choices, but also in the computation of confidence. To formalize this insight, we extended the LPR model[25], such that it also includes confidence: In the c-LPR, we assumed that both choices and confidence are based on the same source of evidence, but confidence represents a further degraded version of the decision variable (DV) relative to a fixed set of confidence criteria. Results of a formal model comparison show that the source of the RIE effect in confidence can be traced back to the computation of the DV, in other words, when transforming hue values using a sigmoidal function before averaging them into a single decision variable. This transformation accentuates differences that are close to the decision bound, leading to increased

**Fig. 3 | Color context flips the response-incongruent evidence (RIE) effect. A** Schematic illustration of the stimuli used for Dataset 10, which could on average be blue, purple, or red (overlapping distributions). Participants were presented with these stimuli either in the Blue Context (blue vs. purple trials) or in the Red Context (red vs. purple trials). Importantly, purple stimuli were physically identical in both contexts as indicated by the example stimuli shown spread across the color spectrum. **B** Regression coefficients for the 8 sorted, colored elements of the identical purple stimuli shown either in the context of red stimuli or in the context of blue stimuli, i.e., depending on the color context. The purple example stimulus from (**A**) is shown below the x-axis, but it should be noted that the regression coefficients shown are based on all purple stimuli from the experiment. Uncertainty is reflected in the error bars (2.5% and 97.5% Wald confidence intervals; $N = 4771$ observations for purple trials in the Blue Context, $N = 4755$ observations for purple trials in the Red Context).

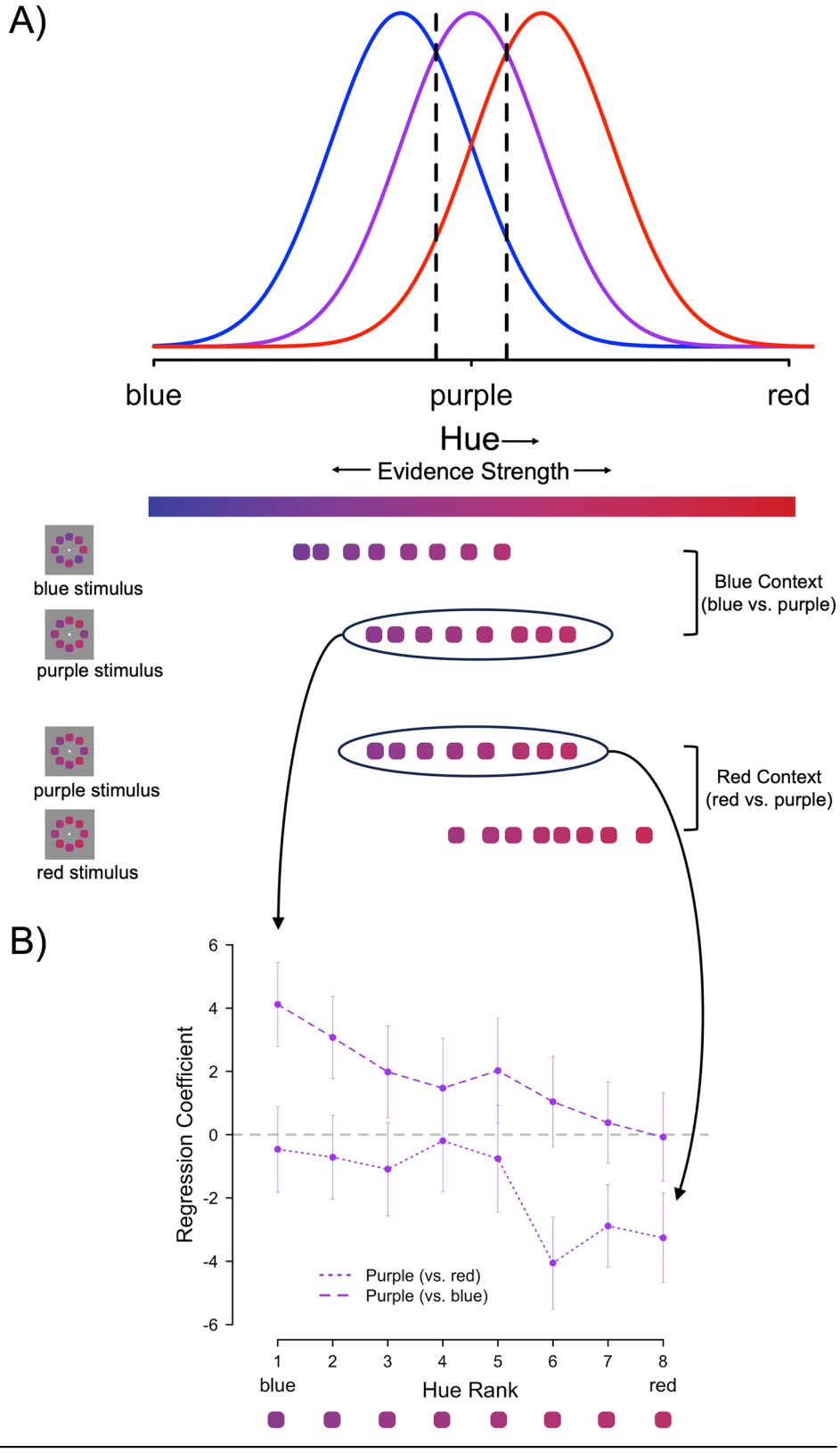

sensitivity at the category boundary and thus an increased weight of incongruent evidence in the computation of confidence. In a final step, we then tested this prediction made by the model in a preregistered experiment. We found that participants were able to flexibly move the category boundary in response to task instruction and that the elements from which identical stimuli were composed contributed differently to confidence depending on the color context in which they were presented. Together, our findings offer an important extension of the literature regarding the computations underlying decision confidence, showing that in the case of continuous evidence spaces, confidence can predominantly be driven by decision-incongruent evidence due to trickle-down effects from the decision formation.

## Confirmation bias meets its counterpart: reconciling the RIE with the RCE

The RCE effect in confidence (also referred to as the Positive Evidence Bias; PEB) is often interpreted as a form of confirmation bias. Confirmation bias is a general cognitive bias that can occur in various domains, including beliefs, opinions, decisions, and social judgments. It refers to the tendency to seek, interpret, and favor information that confirms pre-existing beliefs or expectations, and to ignore or dismiss information that contradicts them[52–54]. In line with this, the RCE effect in confidence has been interpreted as a specific form of confirmation bias because it refers to the tendency to place more weight on decision-confirming evidence when evaluating one's own performance or abilities, while neglecting or underweighting decision-disconfirming evidence[10]. Despite the widespread occurrence of the confirmation bias and the RCE effect, which has been independently replicated across various labs using different behavioral tasks[10–20], the current study has yielded the opposite effect, namely that confidence judgments are more sensitive to RIE. This raises the question why our results differ so drastically. To reconcile these apparent differences, we suggest that the RCE and the RIE effects are caused by two independent mechanisms that occur depending on the number of evidence sources. A useful way to think about this is within an SDT framework. In the case of a one-dimensional SDT decision, like in our task, evidence categories (e.g., 'red' and 'blue' stimuli) can be linked back to a single latent source (here: 'color') and samples are distributed continuously around a category boundary that is located somewhere between the prototypes of these two categories. In our case the boundary lies at purple, equidistant between blue and red. In such cases, it makes sense to have increased sensitivity around this bound as this is where statistically a large mass of the evidence in the environment would occur, leading to the RIE effect[27].

In the case of a two-dimensional SDT decision, like the face versus house task in the study by Peters and colleagues[17], stimuli are composed of two sources of evidence which are mutually exclusive (e.g., a face does not have a roof). This critically alters the decision criterion, which is no longer placed somewhere on the continuum of a single evidence source but is instead placed such that it compares evidence for each option at each level of intensity. In such cases, participants could be able to disregard evidence from the respective other source of evidence (e.g., they can more easily ignore a roof if they believe they are seeing a face), resulting in the RCE effect. Support for this interpretation comes from a study on perceptual estimation by Gershman and Niv[55]: Participants had to estimate the number of circles presented on screen and over time learn the mean of the underlying distributions from which these were drawn. The authors showed that if the two generating distributions overlapped too much, participants assumed all samples were coming from a single category located at the mean between the two distributions. Applied to our situation, this suggests participants need sufficient differences in the perceptual samples to be able to attribute them to distinct latent sources and ignore the irrelevant category (giving rise to an RCE effect). If, however, participants perceive the evidence to come from a single distribution, the robust averaging principle will play out (giving rise to an RIE effect). In line with this proposal, recent work[56] has shown that turning a two-alternative forced choice (2AFC) task into a one-alternative forced choice (1AFC) task—that is by going from discrimination to detection—flips the positive-evidence bias into a 'negative evidence bias'. It remains to be seen whether these findings can be accounted for by the currently proposed c-LPR model implementing the robust averaging principle.

Taken together, this paints the picture of two independent mechanisms giving rise to the RCE and RIE effects. To advance our understanding further, we propose distinct temporal loci for these mechanisms. We have shown that the RIE effect originates from a perceptual, adaptive gain-based principle. The early occurrence of this effect in the processing hierarchy, before any summary statistics are computed, aligns with research indicating that a set's gist is represented without encoding for individual item representations[57]. In contrast, we posit that the RCE effect resides later in the process. Specifically, during the decision phase, there is a selective suppression of incongruent information, promoting self-consistency. Despite these differences, we also need to point out an important commonality: Both effects show that confidence is not an unspecific read-out of the available evidence as some normative models assume (cf. SDT frameworks[6,7]) but instead follows flexible, situational-specific weighting functions.

## Computational modeling results

Our study highlights the importance of computational modeling in revealing the underlying mechanisms that give rise to persistent cognitive biases. However, our work is not the first attempt to describe how pieces of evidence are combined into the computation of confidence, such as the RIE and RCE effect. Previous models can be roughly categorized into two groups. The first group consists of models that hard-code different mechanisms for choices and confidence into the model's architecture[10,13] (for a comparison of different models see Shekhar and Rahnev[58]). These models explicitly assume that only stimulus-congruent evidence drives confidence, similar to our own implementation of the RCE effect (Model 7). A recent biologically inspired model by Maniscalco and colleagues[19] takes this assumption further by linking the effect to a specific type of neuron that varies in how much tuned normalization they display. Reliance on RCE emerges within this model because units with higher tuned inhibition are set to drive decisions, while units with lower tuned inhibition are set to drive confidence. Because the RCE effect is hard-coded in these proposals, it is difficult to see how such a model could jointly account for RCE and RIE effects.

In contrast, the second group of models assumes that choices and confidence arise from the same underlying evidence. Using a deep neural network model that classified noisy visual stimuli and provided a confidence rating, Webb and colleagues[22] showed that the RCE effect naturally arises due to the environmental statistics of the training data. Critically, by manipulating the signal, the noise or both during the training of the model, the RCE reliance could be made to disappear (see also Weise et al.[59]; Shekhar and Rahnev[58]), and even revert: as soon as the model was trained with data in which only the 'noise' dimension varied, the decision-incongruent evidence drove confidence resulting in an RIE effect. The key takeaway from Webb and colleagues[22] is therefore which evidence feeds into confidence judgments is learned, depending on the statistical regularities in the current environmental. Drawing parallels between these earlier findings[22] and our claims that human perception of evidence depends on whether it is perceived to come from one or two latent sources, an interesting similarity emerges. In both scenarios, humans are assumed to adapt flexibly to environmental regularities, whether it is the number of evidence sources or the meaningful variation within these sources. Future studies are needed to put these assumptions to the test. Lastly, a model by Khalvati and colleagues[18] should be mentioned, which equally assumes choices and confidence are computed using the same Bayes-optimal process. In this model though, the RCE effect arises as an artefact merely due to the experimenter's lack of insight into how much evidence the subject has taken into account. Taken together, our model is in line with the second group of models, which posits that choices and confidence are derived from the same underlying decision variable.

As a related question, an increasing number of studies use computational models to unravel how the computation of confidence unfolds over time. A key debate in this domain is whether confidence is modeled as arising only after a choice has been made[60–63] or whether confidence can already be traced back to the initial stage of evidence accumulation informing the choice[64–68]. Although we did not implement a dynamic model in the current work, at first glance our findings appear to be in line with accounts positing that confidence can be traced back to the initial evidence accumulation stage. The observation that both choices and confidence depend on the same compressed evidence suggests that they are based on the same decision variable (i.e., representing the same evidence). Proponents of post-decision models, however, might argue that their models can still account for these data by assuming that confidence is based on a purely post-

decision accumulation process which samples from the same compressed evidence that the choices were based on. Ultimately, implementing and comparing the above models would be needed to answer this question.

## Limitations

While the present study offers insights into the role of response-incongruent evidence in confidence judgments, several limitations should be noted. First, our analyses were restricted to color categorization tasks, leaving open whether the observed effects generalize to other modalities or decision contexts. Second, although our computational modeling captured key aspects of the data, we did not implement dynamic models of evidence accumulation, limiting the conclusions about the temporal dynamics of confidence computation. Third, future work will need to establish a direct link between the RIE effect observed here and the RCE/positive evidence bias within the same participants to clarify the mechanisms driving these biases.

## Future directions

A promising future avenue is investigating the RIE effect in other modalities. In the current study, we only used color stimuli, leaving open the possibility that the observed robust averaging effect and RIE only exist for color perception. Animal studies have long suggested that color is represented in both a linear and nonlinear fashion throughout the visual cortex[69]. Moreover, color perception near a category boundary in humans has been suggested to follow a nonlinear (sigmoid) function[70]. It is unclear whether other modalities that are predominantly represented in a linear fashion would show robust averaging and a corresponding RIE effect. However, it should be mentioned that de Gardelle and Summerfield[25] also found the robust averaging effect for their task in a condition where shape averaging was required. The same held true for the study by Li and colleagues[28] who used orientation stimuli. While those earlier studies only focused on choices, it is reasonable to assume the same for confidence calculation.

## Conclusion

In summary, we re-analyzed nine color categorization datasets and found reliance on response incongruent evidence (RIE) in confidence. This finding contradicts both SDT models of decision confidence assuming an equal readout of evidence and studies reporting RCE reliance. Using computational modeling, we show that our effect can be explained by heightened sensitivity around the category boundary, an interpretation for which we find direct empirical support. Our findings show that in the case of continuous evidence spaces, confidence is predominantly driven by decision-incongruent evidence due to downstream effects from the decision formation.

## Data availability

All data is available for download at https://osf.io/4326x/files/osfstorage/68cd0bca8b8c350fb31e46f0.

## Code availability

All code is available for download at https://osf.io/4326x/files/osfstorage/68cd0e04c55a5cf9be3b7e50 (https://doi.org/10.17605/OSF.IO/4326X).

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

## Acknowledgements

This research was funded in part by the Wellcome Trust (Sir Henry Wellcome Postdoctoral Fellowship 206480/Z/17/Z awarded to A.B.). The funder had no role in study design, data collection and analysis, decision to publish or preparation of the manuscript. For the purpose of open access, the author has applied a CC BY public copyright license to any Author Accepted Manuscript version arising from this submission. The authors thank their collaborators from past studies for that allowed pooling of such a large number of datasets. A special thanks to Nick Yeung, whose thoughtful supervision and consistent guidance enriched these studies. Furthermore, we would like to thank Eline Van Geert, Megan Peters, Brian Maniscalco, Marco Wittmann, Sam Gilbert, and Tom Verguts for useful discussions. Many thanks also to Chris Summerfield and Vincent de Gardelle for providing the LPR model code as a starting point for the c-LPR model.

## Author contributions

Annika Boldt: Conceptualization, methodology, software, validation, formal analysis, investigation, resources, data curation, writing—original draft preparation, writing—review & editing, visualization, supervision, project administration, funding acquisition. Yishu Sun: Formal analysis, investigation, writing—review & editing. Kobe Desender: Conceptualization, methodology, software, validation, formal analysis, investigation, resources, data curation, writing—original draft preparation, writing—review & editing, visualization, supervision, project administration, funding acquisition.

## Competing interests

The authors declare no competing interests.
