## [Transparent Peer Review file · Communications Psychology]

How disconfirmatory evidence shapes confidence in decision-making

Corresponding Author: Dr Annika Boldt

Version 0:

Decision Letter:

Dear Dr Boldt,

Thank you for your patience during the peer-review process. Your manuscript titled "Dis-confirmatory evidence drives confidence" has now been seen by 2 reviewers, and I include their comments at the end of this message. They find your work of interest but raised some important points. We are interested in the possibility of publishing your study in Communications Psychology, but would like to consider your responses to these concerns and assess a revised manuscript before we make a final decision on publication.

We therefore invite you to revise and resubmit your manuscript, along with a point-by-point response to the reviewers. Please highlight all changes in the manuscript text file.

Editorially, we consider it important that the revised manuscript include additional analyses requested by the reviewers, in particular, please test a PEB model that increases (or decreases) the contribution of the choice congruent evidence, assessment of individual variability/differences in the findings, and report whether the difference in AIC of the models is statistically different.

Please ensure you follow our statistical guidelines when reporting statistics (<https://www.nature.com/commspsychol/submit/submission-guidelines#statistical-guidelines>). Please note in particular our requirements for the reporting and interpretation of null-results. Non-significant findings derived from null-hypotheses significance tests should be reported in full, but may not be interpreted. Where you interpret null results, this interpretation must be based on Bayes Factors or equivalence tests.

I am attaching an Editorial Requests Table that details critical reporting requirements for the revised manuscript. Please attend to each item and ensure your manuscript is fully compliant. We are requesting that your manuscript aligns with these requirements as this facilitates the evaluation of your manuscript, reducing delays in re-review and potential future acceptance. If your revised manuscript is not aligned with these requests on major issues, such as those concerning statistics, it may be returned to you for further revisions without re-review. Additional information can be found in our style and formatting guide <https://www.nature.com/documents/commspsychol-style-formatting-guide-accept.pdf> Communications Psychology formatting guide.

Please use the following link to submit your

- revised manuscript,
- point-by-point response to the referees' comments,
- cover letter (as a separate document),
- the Editorial Policy Checklist (see below),
- the Reporting Summary (see below), and
- the completed Editorial Request Table (attached):

Link Redacted

Best regards,

Jennifer Bellingtier

Jennifer Bellingtier, PhD
Senior Editor
Communications Psychology

REVIEWER EXPERTISE:

Reviewer #1 decision making, metacognition, computational models

Reviewer #2 decision making, metacognition, computational models

REVIEWER REPORTS:

Reviewer #1 (Remarks to the Author):

In this manuscript, the authors reanalysed published behavioral datasets consisting of a color discrimination task with a confidence rating. They found that confidence levels were better explained by the level of disconfirmatory evidence, a finding that differs from some previous studies showing that confidence is driven by confirmatory evidence. Using computational modelling they then show that decisional information corrupted by noise is sufficient to explain this effect. Finally, they conducted a pre-registered study to confirm that the context could modulate the type of evidence correlating with confidence.

Understanding the mechanisms underlying confidence is important and this study adds an interesting piece to the puzzle. The modelling part and the statistics are very well conducted and the additional pre-registered study adds value to the whole. The manuscript is well written (although a few parts can be difficult to understand, probably due to a will of being concise). Although the authors do propose how this mechanism could extend to other tasks where the positive evidence bias was found they did not directly test it, which would have strengthened the findings. Still, I think that this work would make a very nice contribution to the field, but I suggest that the authors reconsider some aspects of the modelling.

1) It seems quite trivial that the no-compression models (3, 5 and 6) will not reproduce the RIE for choice in Figure 2. Therefore, I do not think that they make a very good job at supporting the authors claim in the model comparison. Instead, I would be more interested in seeing alternative mechanisms for confidence. The authors tested a model scaling of the metacognitive noise based on stimulus variance (model 1) which makes sense, and fixing the metanoise to a somewhat arbitrary level of the decisional noise (model 4).

However, I was surprised that the authors did not test a PEB model that increases (or decreases) the contribution of the choice congruent evidence. From Figure 2, it actually seems that model 2 is largely overestimating the choice congruent evidence going against the PEB but also against the authors' claim that the response congruent effect is driven by the same decisional evidence.

2) Considering the large number of participants, it would seem easy to report whether the difference in AIC of the models is statistically different. Otherwise, the claims that can be done on the winning model are limited.

3) I understand the need to present data averaged across participants but it would be nice to see whether this simple 3 parameters model captures individual variability in the behavior, or whether it can only describe average behavior.

4) I was surprised by the title of the manuscript. according to what I understood, the fact that disconfirmatory evidence drives confidence is an artifact coming from sampling confidence from a decisional variable compressing sensory evidence. Or if I am mistaken, then what is the parameter in the model that would explain the disconfirmatory part?? This made me think that

it is not necessarily clear in the manuscript how this effect actually arises. It would be very informative to have a supplementary figure showing how the different decisional and confidence parameters affect RIE confidence (and to a less extent RIE choice).

Minor:

70 - 73 -> It is unclear to me how "ignoring response incongruent evidence can inflate confidence". Almost every model of confidence has a bias factor that can account for under or overconfidence.

117 - 124; The hypotheses were not clear to me.

Is the SEM computed across participants or across datasets? I assume it is computed across datasets? Why is there no SEM on Figure 3B?

It is a bit hard to grasp what the 'context' of the pre-registered experiment is. qs

Reviewer #2 (Remarks to the Author):

The paper explores how dis-confirmatory evidence influences confidence in decision-making. Contrary to the normative model, which suggests equal weighting of evidence, and the positive-evidence bias (PEB) model, which suggests an overweighting of choice-confirming information, this study finds that people give more weight to evidence conflicting with their initial choices. The authors re-analyzed nine datasets and conducted a new preregistered experiment to show that confidence relies more on decision-incongruent evidence due to the robust averaging principle, which places greater weight on evidence close to the category boundary. The authors also discuss the potential reason for the inconsistency between the current work and the previously found positive evidence bias, suggesting that it might be due to the difference between one-dimensional versus two-dimensional signal detection decisions.

I think this paper is overall strong, as it includes a large sample size and the results are robust and interesting. The authors' modeling work, in addition to the behavioral results, provides strong evidence supporting their findings. Additionally, the authors propose possible mechanisms explaining the discrepancy between the current results and previous literature. The hypothesis and conclusions are clear and easy to understand, and the results are important for anyone interested in confidence judgments. Overall, I believe that this paper advances the field of metacognition. I only have one major comment:

Pooling data together, either for subjects or across experiments, hides potentially interesting patterns. While the authors aim to examine an overall effect of whether dis-confirmatory evidence drives confidence more than confirmatory evidence (the response-incongruent effect), potential individual differences would be interesting to explore and could advance the understanding of this effect. The same applies to across-study differences, though I imagine there isn't much heterogeneity across studies since the tasks used in all 10 experiments are essentially the same.

Below are my minor comments. Overall, they are mostly suggestions, and the authors might choose whether or not to implement them.

1. Color perception varies among individuals, affecting their category boundary for differentiating blue vs. red. This factor is related to my major comment, and I wonder how/whether it is incorporated into the results.
2. In Figure 1C, the y-axis label "Average Hue" has an arrow pointing up, and Figure 1B also has an arrow next to the label "Hue." More explanation is needed for Figure 1C.
3. Were the subjects incentivized in any way to accurately use their confidence rating?
4. Is there a change of strategy over time? Analyzing the data split into halves might reveal if reliance on choice-incongruent evidence changes throughout the experiment, as training could affect confidence computation. For instance, Buonocore and Riccaboni (2015) found that metacognitive training can reduce the bias against dis-confirmatory evidence in patients with schizophrenia.
5. Discuss how the absence of feedback in their experimental design might affect their results would potentially be interesting. Feedback has been shown to reduce bias (Haddara & Rahnev, 2022).
6. In the sanity check where conditions differ in confidence but not accuracy, can the authors provide the statistical comparison for confidence? It is currently missing from the Results.
7. Mentioning the six model variants earlier would clarify the context when Models 2 and 3 appear in Figure 2. It currently seems abrupt without prior introduction.
8. The trend reversal of regression coefficients in positions 3 and 6 for purple (vs. red) and in position 4 for purple (vs. blue) should be discussed.
9. Additional thoughts on the models of confidence and decisions based on the same evidence could enhance the discussion. There are strong post-decisional models of confidence (e.g., Pleskac & Busemeyer, 2010; Moran et al., 2015; Herregods et al., 2023) and models suggesting confidence signals during initial evidence accumulation (e.g., Dotan et al., 2018; Hellmann et al., 2022; Rahnev et al., 2016; Ratcliff & Starns, 2009; Vickers, 1979; Yu et al., 2015). While the authors cite Shekhar & Rahnev (2022), expanding on these models could be beneficial.
10. The paper Shekhar & Rahnev (2022) is now published at Journal of Experimental Psychology: General. DOI number: 10.1037/xge0001524.

EDITORIAL POLICIES

We ask that you ensure your manuscript complies with our editorial policies and reporting requirements.

To that end, we require revised manuscripts to be accompanied by two completed items: a reporting summary that collects information on study design and procedure, and an editorial policy checklist that verifies compliance with all required editorial policies.

- <https://www.nature.com/documents/nr-reporting-summary.zip>>Nature Research Reporting Summary
- <https://www.nature.com/documents/nr-editorial-policy-checklist.pdf>>Editorial Policy Checklist

All points on the policy checklist must be addressed. Your revised manuscript can only be sent back to the referees if these checklists are completed and uploaded with the revision.

Notes: If you have submitted a Stage 1 Registered Report, Review, Primer, Comment, or Perspective you do not need to submit these forms. If you have already submitted these forms, you may disregard this request.

** Visit Nature Research's author and referees' website at <http://www.nature.com/authors>>www.nature.com/authors for information about policies, services and author benefits**

If you experience problems in linking your ORCID, please contact the <http://platformsupport.nature.com/>>Platform Support Helpdesk.

Version 1:

Decision Letter:

Dear Dr Boldt,

Your manuscript titled "How Disconfirmatory Evidence Shapes Confidence: Downstream Consequences of Decision-Making" has now been seen by our reviewers, whose comments appear below. In light of their advice I am delighted to say that we are happy, in principle, to publish a suitably revised version in Communications Psychology.

We therefore invite you to revise your paper one last time to address the remaining concerns of our reviewers and a list of editorial requests. At the same time we ask that you edit your manuscript to comply with our format requirements and to maximise the accessibility and therefore the impact of your work.

EDITORIAL REQUESTS:

SUBMISSION INFORMATION:

OPEN ACCESS:

* DATA AVAILABILITY:

Link Redacted

Best regards,

Jennifer Bellingtier

Jennifer Bellingtier, PhD
Senior Editor
Communications Psychology

REVIEWERS' EXPERTISE:

Reviewer #1 decision making, metacognition, computational models
Reviewer #2 decision making, metacognition, computational models

REVIEWERS' COMMENTS:

Reviewer #1 (Remarks to the Author):

The authors have answered all my comments/concerns but one.

Reviewer 2 and I had the same request (R1_3 and R2_1): to see whether the model accounts for individual differences instead of data averaged across participants. I don't think that Fig. S2 addresses our point about individual differences (it

shows differences across datasets, not across participants). So I assume that the model does not reproduce individual participants' behavior. Since this is a limitation but does not invalidate the results, and to avoid further delaying the manuscript's acceptance, I propose that the authors simply acknowledge this in the manuscript.

Thank you for testing a new model for the PEB, I found that it makes the paper stronger.

Congratulations on a nice paper.

Reviewer #2 (Remarks to the Author):

I appreciate the authors for addressing all comments. I have no further questions.

Reviewer 1

RI_1) "It seems quite trivial that the no-compression models (3, 5 and 6) will not reproduce the RIE for choice in Figure 2. Therefore, I do not think that they make a very good job at supporting the authors claim in the model comparison. Instead, I would be more interested in seeing alternative mechanisms for confidence. The authors tested a model scaling of the metacognitive noise based on stimulus variance (model 1) which makes sense, and fixing the metanoise to a somewhat arbitrary level of the decisional noise (model 4). However, I was surprised that the authors did not test a PEB model that increases (or decreases) the contribution of the choice congruent evidence. From Figure 2, it actually seems that model 2 is largely overestimating the choice congruent evidence going against the PEB but also against the authors' claim that the response congruent effect is driven by the same decisional evidence."

As suggested by the reviewer, we have tested whether a PEB model can account for our findings. Before going into the details, we would like to point out that we identified a mistake in the calculation of the AIC in the previous version of our manuscript. As previously mentioned in Equation 6 (now called Equation 9), we previously used:

$$AIC = 2k + 2n * \ln(SSE/n)$$

Whereas the correct formula (now corrected in the current version of the manuscript, Equation 9 on p. 34) is:

$$AIC = 2k + n * \ln(SSE/n)$$

This mistake under-punished models with more free parameters, putting more weight on the SSE. After (re)calculating the AICs in the appropriate way, the winning model out of the six variants tested previously now was Model 4 instead of Model 2 reported previously. Note that these models only differed in whether metanoise was a free parameter (or fixed to the same level as decnoise) so essentially this shows that it is not worth sacrificing a free parameter to freely estimate metanoise. Fortunately, our conclusions about the models without sigmoidal compression were unaffected, as all three model variants including a linear transformation function provided a considerably worse fit. We apologize for this mistake, which we have corrected in the new version of our manuscript.

Next, we turn towards the PEB model. In the literature, the PEB model is typically implemented as follows (Maniscalco et al., 2016; Shekhar & Rahnev, 2023):

$$DV1=N(u1,SD)$$

$$DV2=N(u2,SD)$$

$$\text{Choice} = \text{sign}(DV1-DV2)$$

$$\text{Confidence} = \{ f(DV1), \text{choice} = u1; f(DV2), \text{choice} = u2 \}$$

In the context of our experiment, we thus implemented a model where confidence is computed based on response-congruent evidence only. This model had the same number of free parameters as Model 4 (i.e., the winning model) with the only difference being the way that confidence was quantified. Given that this model only considered choice-congruent evidence to compute confidence, it was necessary to add decision-noise to individual elements instead of adding

decision-noise to the group mean (as done for the other models). Thus, for the PE model we added noise to the individual elements and only then computed the mean of those eight (noisy) elements:

$$E_i = C_i + N(0, \varepsilon_1)$$

To compute choices, we transformed these values using the same compression as in the other models and then computed the mean

$$DV_1 = \frac{1}{8} * \sum_{i=1}^8 f(E)$$

For confidence, however, we computed the decision variable as

$$DV_2 = \sum E_{congruent} + N(0, \varepsilon_2)$$

Where $E_{congruent}$ reflect choice-congruent elements.

As can be seen in the table below, Model 4 was still the preferred model based on both mean AIC and sum AIC. Most importantly for the current purpose, Model 7 (PE) performed very poorly, suggesting that participants do not compute confidence based on response-congruent evidence.

*Table 1. We compared seven different models in their ability to fit average accuracy and average confidence in each of the cells of the design. The winning model (Model 4) is highlighted in bold. Note that for Models 4-7 we did assume metacognitive noise to be present (indicated by *), but its value was fixed to be the same value as decnoise. Model 7 incorporates reliance of response-congruent evidence (RCE).*

Parameter									
Model	compression	compression	decnoise	metanoise	scalemetanoise	hardcoded RCE	df	M(AIC)	SUM(AIC)
1	✓	-	✓	✓	✓	-	4	-35.23	-6201
2	✓	-	✓	✓	-	-	3	-36.65	-6451
3	-	✓	✓	✓	-	-	3	-33.58	-5911
4	✓	-	✓	*	-	-	2	-37.30	-6564
5	-	✓	✓	*	✓	-	3	-34.16	-6011
6	-	✓	✓	*	-	-	2	-34.48	-6069
7	✓	-	✓	*	-	✓	2	-28.11	-4947

Moreover, Model 7 also failed to capture the RIE effect. As can be seen in the updated Figure S1 below:

F) Model Recovery

G) Modelled Choices

H) Modelled Confidence

Figure S1. Full behavioral and model findings. Columns A-E depict the empirical lines and simulated bands for A) **accuracy** by condition, B) **confidence** by condition, C) confidence frequency by accuracy, D) choice reliance effects and E) confidence reliance effects. Uncertainty for the empirical data is reflected in the error bars (2.5% and 97.5% Wald confidence intervals). Uncertainty for the simulated data is reflected in the width of the bands (2.5% and 97.5% Wald confidence intervals). We collapsed across datasets as much as possible. Each row depicts the results for one model using the best-fitting parameter combinations for this respective candidate model. F) Model recovery matrix, showing the probability that data generated under a specific model is best accounted for by that same model (diagonal) versus the other models under consideration (off-diagonal). G-H) show the confidence extension of the log posterior ratio model (c-LPR) using data for the winning model (Model 4) with parameters from a single subject from Dataset 1. G) Stimuli in the experiment were sampled pseudorandomly, meaning that the average colors of the individual stimuli were kept in narrow tolerance bands around the 4 condition mean hue (x-axis). The y-axis depicts the mean decision variable resulting from robust averaging. Dashed horizontal lines reflect the confidence criteria for blue and red stimuli. The **black** dot reflects the stimulus example shown in Figure 2C). H) Transformation of the mean decision variable into confidence by means of the respective confidence criteria (blue and red horizontal dashed lines).

We have now added this additional model to Table 1 in the manuscript together with its description in the following places in the text.

Introduction on p. 7:

“Such a model furthermore outperformed a competing model that implements the positive evidence bias (RCE effect).”

Results on pp. 9-10:

“The lines in Figure 2A – here superimposed onto simulations generated from two out of seven models further described below – show the resulting parameters from two logistic mixed models fitted to all pooled data predicting choices based on the hue values of the eight colored elements of each stimulus, separately for each color category (i.e., red or blue).”

Updated Figure 2:

Model 3

A) RIE Choice

B) RIE Confidence

Model 4

C) Robust Averaging

Figure 2. Behavioral and simulated reliance (RIE) effects. **A)** Choice and **B)** confidence effects as a function of stimulus category (red or blue). The top row depicts the results for Model 3, the bottom row for Model 4 (the winning model), each using the best-fitting parameter combinations for this respective candidate model. The stimulus-composing elements were ranked by their hue and the color for each of the resulting eight ranks used as a predictor for choice (A) or confidence (B) respectively. The resulting slopes of the regression coefficients indicate that both choices and confidence judgements were relatively more affected by response-incongruent evidence (RIE). In other words, for a red stimulus, the bluest elements showed the strongest effect and vice versa. The empirical data is depicted as lines with uncertainty reflected in the error bars (2.5% and 97.5% Wald confidence intervals). The simulated data is depicted as bands with uncertainty reflected in the width of the bands (2.5% and 97.5% Wald confidence intervals). Whether the resulting regression weights are positive or negative depends entirely on how they were entered into the regression and is arbitrary. The focus lies on their relative contributions in the prediction. **C)** The c-LPR model assumes robust averaging, meaning that instead of directly averaging the hues of which the stimulus is composed (in our experiment hue values of the individual elements), these first get passed through a sigmoidal transfer function. Such a transfer leads to a compression of differences at the extremes (red and blue items) and augments differences at the category boundary (purple items).

Results on p. 15:

“As can be seen, there are two strongly red elements, 0.15 and 0.19, which are at a sizable distance from each other in hue space, but after the sigmoidal transfer they are mapped onto 0.990 and 0.998 in DV space. As a consequence, variation in these elements is effectively abolished. Now compare this to two elements that lie close to the boundary with a hue value of -0.01 and 0.03. After the sigmoidal transfer these are mapped onto -0.24 and 0.52 in DV space, and thus their difference has been artificially increased.”

Results on pp. 16-17:

“Table 1 lists the seven different competing models that we tested in the current work. Model 4 was the simplest model that we considered, which was essentially the LPR model with the addition of confidence as a further degraded variant of the decision variable *DV* (2 free parameters: compression and decnoise). Note that in this model the amount of metacognitive noise that was added to the signal before reading out confidence was assumed to be the same as the amount of decision noise (i.e., controlled by decnoise). Model 1 was the most complex model we considered, which features all four free parameters just described (4 free parameters: compression, decnoise, metanoise, scalemetanoise). In Model 2, we dropped the assumption of scalemetanoise (i.e., setting its value to zero), but did estimate metanoise. In Model 3, Model 5, and Model 6 we implemented a linear (rather than a sigmoidal) compression parameter – implying that hues are linearly mapped onto the [-1, 1] range before being averaged into the DV. Finally, in Model 7 we implemented the positive evidence bias, or RCE effect. To calculate confidence, each individual element is first distorted by noise. The model then discards all decision-incongruent elements and calculates confidence based on this raw response-congruent stimulus evidence (i.e., untransformed

evidence). These changes only affect confidence, whereas choices continue to be based on compressed evidence. These models allow to directly test the importance of the compression parameter in explaining the RIE effect in choice and confidence.”

Results on pp. 17-18:

“Model 4 (featuring compression and denoise as free parameters) provides the best fit at a quantitative level, with a difference in average AIC of 0.64 and summed AIC of 113 compared to the second-best model (Model 2). When inspecting these quantitative results, it becomes clear that models without the compression parameter perform worse (Δ AICs > 2.81 for average, and Δ AICs > 495 for summed) compared to Model 4.”

Results on pp. 18-19:

“To inspect this possibility, we simulated data from the different models and compared model predictions to the patterns seen in the empirical data (see Figure 2A-B for Models 3 and 4, and Figure S1D-E for all models). In particular, Figure 2B shows that Model 4 closely captures the critical finding of the current manuscript; both in the data and in the model fit we see that confidence is mostly computed based on stimulus-incongruent evidence, whereas variation in stimulus-congruent evidence does not bear much weight. Indeed, when applying the same contrasts to the results of the mixed model analysis previously performed on the empirical data to data simulated under Model 4, we observed negative slopes for red and positive for blue stimuli, p s < .001, stronger weights for incongruent versus congruent elements both for blue and red stimuli, p s < .001, and stronger weights for inlying than for outlying elements both for blue and red stimuli, p s < .001. At this point, we want to stress again that the fact that Model 4 reproduces the RIE effect is far from trivial as the models were merely optimized to fit average accuracy and average confidence; they were not fit to the regression data. Rather, the regression lines (i.e., the RIE effect) naturally fall out of Model 4. For completeness, the estimates of the free parameters in Model 4 can be found in the SI Appendix in Figure S5A.

Inspection of the different models (cf. Figure S1) reveals that the compression parameter is key in explaining the reliance effect. The three models that do not contain the compression parameter (i.e., Models 3, 5, and 6) not only provide a substantially lower quantitative fit, also qualitatively these three models completely miss the reliance effect in confidence; further demonstrating the importance of robust averaging in explaining the reliance effect in confidence.”

Results on pp. 19-20:

“Interestingly, although the model comparison favors Model 4 over the other models in terms of quantitative fit, it does seem to fall short in explaining the medium-condition effect. Indeed, while the predictions from Model 4 show that the high mean, high variance and the low mean, low variance conditions were not significantly different in terms of accuracy, $p = .123$, $BF = .40$, they were also not different from each other in terms of confidence, $p = .404$, $BF = 0.17$. Further inspection of the different models reveals that only the models including scalemetanoise can

explain this dissociation. Indeed, simulations from Model 1 mirrored the empirical finding that accuracy does not significantly differ between the two medium conditions, $p = .59$, $BF = .34$, whereas it does predict a significant difference in confidence, $\chi^2(1) = 8.05$, $p = .005$, $BF = 601926$. Thus, our modeling efforts explain the medium-condition effect as a selective influence of stimulus variance on metanoise *over and above* its effect on decision noise. Note, however, that this effect was rather subtle and therefore at the quantitative level Model 4 won the competition despite not being able to capture this effect.”

Results on p. 20:

“*No flexible metanoise parameter*

A final take-away of our modeling efforts is that our data favored a model assuming that metanoise takes the same value as decnoise, such as in the winning model (Model 4). Models including a separate (i.e., flexible) parameter to control for metanoise (Models 1-3) perform worse. Several recent studies have investigated whether noise at the choice and confidence level could be dissociated (e.g., Balsdon, Wyart & Mamassian, 2020; Xue, Shekhar & Rahnev, 2021). Our findings speak against such an additional noise source.”

Discussion on p. 26:

“These models explicitly assume that only stimulus-congruent evidence drives confidence, similar to our own implementation of the RCE effect (Model 7).”

Methods on pp. 33-34:

“With Model 7, we further chose to implement the RCE effect, or PEB. Specifically, we implemented a model where confidence is computed based on response-congruent evidence only. This model had the same two free parameters as Model 4 (i.e., the winning model) with the only difference being the way that confidence was quantified. Given that this model only considered choice-congruent evidence to compute confidence, it was necessary to add decision-noise to individual elements instead of to the group mean (as done for the other models). Thus, for the PEB model we first added noise to the individual elements:

$$E_i = C_i + N(0, \varepsilon_1) \quad (5)$$

Then, we transformed these values using the same compression function, f , as in the other models (i.e., Equation 1) and then computed the mean:

$$DV_1 = \frac{1}{8} * \sum_{i=1}^8 f(E) \quad (6)$$

For confidence, however, we computed the decision variable as

$$DV2 = \sum E_{congruent} + N(0, \varepsilon_2) \quad (7)$$

Where $E_{congruent}$ reflect choice-congruent elements.

Finally, confidence for all models was calculated on a six-point scale by comparing DV_2 to a fixed set of decision criteria. If x equals to 1, the criteria were {Inf; 0.60; 0.35; 0.10; -0.15; -0.40; -Inf}. If x equals to -1, the same criteria were used after multiplying these by -1.”

Methods on pp. 34-35:

“*Parameter recovery.* We simulated data from the winning model for one hundred agents for which we randomly selected a value for each of the two parameters. Subsequently, these data were fitted with the (winning) computational model, and correlations between true and fitted parameters were examined. We randomly selected values from a uniform distribution for compression (range between .00001 and .2) and decnoise (range between .00001 and .3). As a sanity check, we first simulated a large number of trials (8000 trials per simulated agent), which provided excellent recovery for both parameters, $r_s > .99$. We then repeated this process with only 250 trials per simulated agent, corresponding to the experiment with the lowest trial counts. Recovery for the two parameters remained excellent ($r_s > .989$; see Figure S5B).”

We furthermore also updated supplementary Figure S4 (now re-numbered to S5):

Figure S5. Parameters of the winning model (Model 4). *A*) shows parameters distributions for the compression, denoise, and metanoise parameters. The colors indicate the dataset. The superimposed boxplot shows the median and spans from the interquartile range with whiskers indicating the minimum and maximum values respectively and outliers represented as black dots. *B*) Depicts results from the parameter recovery (with $N = 100$ and assuming 250 trials per simulated dataset) for the winning model for the compression, denoise, and metanoise parameters. Simulated values (sim.) are shown on the x-axes and recovered values (rec.) on the y-axes.

R1_2) Considering the large number of participants, it would seem easy to report whether the difference in AIC of the models is statistically different. Otherwise, the claims that can be done on the winning model are limited.

As suggested by the reviewer, we have performed an analysis of differences between AIC indices. Specifically, we predicted AIC indices using a linear mixed effects model with a random slope for subjects, nested within experiments and with model as a fixed effect. Unsurprisingly,

this model showed a main effect of model, $F(6, 1050) = 118.3, p < .001$. More interestingly, we next constructed post-hoc contrasts comparing the winning model (Model 4) to the other models. This showed that Model 4 had significantly lower AIC values compared to models 1, 3, 5, 6 and 7, $ps < .001$, whereas the fit indices were only numerically lower compared to model 2, $p = .0995, BF = 1290$ (although the Bayes Factor strongly favored Model 4). We have now added these statistical results on p. 18:

“We have furthermore performed an analysis of differences between AIC indices. Specifically, we predicted AIC indices using a linear mixed effects model with a random slope for subjects, nested within experiments and with model as a fixed effect. Unsurprisingly, this model showed a main effect of model, $F(6, 1050) = 118.3, p < .001$. More interestingly, we next constructed post-hoc contrasts comparing the winning model (Model 4) to the other models. This showed that Model 4 had significantly lower AIC values compared to models 1, 3, 5, 6 and 7, $ps < .001$, whereas the fit indices were only numerically lower compared to Model 2, $p = .0995, BF = 1290$ (although the Bayes Factor strongly favored Model 4).”

RI_3) I understand the need to present data averaged across participants but it would be nice to see whether this simple 3 parameters model captures individual variability in the behavior, or whether it can only describe average behavior.

We agree with the reviewer that it would be useful for the reader have some information about across-study variability. Therefore, we have included an additional supplementary Figure, which shows that different experiments followed roughly the same pattern. This resulted in the following edits:

Results on p. 18:

“As can be seen in Figure S1A and S1B, most models do a decent job at capturing average accuracy and average confidence in each of the cells of the experiments, and they do so across experiments (Figure S2).”

We have furthermore added the new figure to the SI and renumbered all other figures accordingly:

Figure S2. Full behavioral and model findings for each dataset. Panels A-F depict the empirical lines and simulated bands for A) accuracy by condition, B) confidence by condition, C) choice reliance effects, and D) confidence reliance effects. Uncertainty for the empirical data is reflected in the error bars (2.5% and 97.5% Wald confidence intervals). Uncertainty for the simulated data is reflected in the width of the bands (2.5% and 97.5% Wald confidence intervals). Each experiment is shown in a different color. Only results from the best-fitting model is shown here (Model 4) as colored bands (Panels A and B) or grey bands (Panels C-F). Please note that for Panels C-F, the grey band shows the estimate taken from the mixed models used to fit all data at once and hence there are no individual simulation results for each dataset.

RI_4) "I was suprised by the title of the manuscript. according to what I understood, the fact that disconfirmatory evidence drives confidence is an artifact coming from sampling confidence from a decisional variable compressing sensory evidence. Or if I am mistaken, then what is the parameter in the model that would explain the disconfirmatory part?? This made me think that it is not necessarily clear in the manuscript how this effect actually arises. It would be very informative to have a supplementary figure showing how the different decisional and confidence parameters affect RIE confidence (and to a less extent RIE choice)."

We agree with the reviewer that the title of the manuscript might be confusing. The reviewer is indeed correct that the disconfirmatory part is a downstream consequence (or artifact in the words of the reviewer) from sampling confidence from a DV compressing sensory evidence. Therefore, we decided to revise the title which now reads:

"How Disconfirmatory Evidence Shapes Confidence: Downstream Consequences of Decision-Making"

Regarding the proposed additional supplementary figure, we would like to point the reviewer to Figure S1 where we show the simulated data for each of our models with its best-fitting parameter combination. This should hopefully clear up some of these questions of how the existence (or non-existence) of a given parameter affects RIE confidence/choice.

RI_5) "70 - 73 -> It is unclear to me how "ignoring response incongruent evidence can inflate confidence". Almost every model of confidence has a bias factor that can account for under or overconfidence."

Thanks for raising this point. We now explain more clearly on p. 4 that the inflated confidence in this case does not result from a bias factor (as in other models) but instead is inherently to the processing of the stimulus. Conceptually, this is an important distinction.

"Crucially, in this case such overconfidence would not be due to a bias parameter as some other models might implement it, but is inherent in the evidence processing."

RI_6) “117 - 124; The hypotheses were not clear to me.”

We rewrote these sentences on p. 6 and we hope that the hypotheses are now much clearer.

“Under standard SDT assumptions, it is expected that all elements contribute equally to the computation of confidence. This assumption is shown as the dotted, horizontal line in Figure 1D. If on the other hand, our data were to reveal an RCE effect, this would be reflected in a pattern shown in Figure 1D as the solid line: Under this account, we would expect confidence to mostly depend on the amount of red (vs. blue) in the stimulus when the category was red (vs. blue). As a third option, our data could reveal the inverse effect (dashed lines in Figure 1D). This pattern follows directly from the robust averaging principle: In a stimulus where colored samples are distributed around a red mean, the blueish evidence lies, by definition, closest to the category boundary and vice versa (Figure 1B).”

RI_7) “Is the SEM computed across participants or across datasets? I assume it is computed across datasets? Why is there no SEM on Figure 3B?”

We must apologise. We had incorrectly copied some of the Figure descriptions for Figures 2, S1 and S4 and incorrectly stated that uncertainty was reflected in SEMs. Instead, the error bars or bands in these Figures (and all other one showing results from the mixed model fits) are Wald confidence intervals. We have now added in the manuscript that the confidence intervals are obtained from the mixed model, and so they are computed across participants while taking into account the nesting of participants within datasets. We now specify this in the Methods section on p. 32:

“Wherever confidence intervals for the fixed effects are shown, these are obtained directly from the mixed models. They reflect the estimated population-level effects, computed across participants while accounting for the nesting of participants within datasets through the model’s random-effects structure.”

We have changed the respective figure captions:

“**Figure 2. Behavioral and simulated reliance (RIE) effects.** **A)** Choice and **B)** confidence effects as a function of stimulus category (red or blue). The top row depicts the results for Model 3, the bottom row for Model 4 (the winning model), each using the best-fitting parameter combinations for this respective candidate model. The stimulus-composing elements were ranked by their hue and the color for each of the resulting eight ranks used as a predictor for choice (A) or confidence (B) respectively. The resulting slopes of the regression coefficients indicate that both choices and confidence judgements were relatively more affected by response-incongruent evidence (RIE). In other words, for a red stimulus, the bluest elements showed the strongest effect and vice versa. The empirical data is depicted as lines with uncertainty reflected in the error bars (2.5% and 97.5% Wald confidence intervals). The simulated data is depicted as bands with

uncertainty reflected in the width of the bands (2.5% and 97.5% Wald confidence intervals). Whether the resulting regression weights are positive or negative depends entirely on how they were entered into the regression and is arbitrary. The focus lies on their relative contributions in the prediction. C) The c-LPR model assumes robust averaging, meaning that instead of directly averaging the hues of which the stimulus is composed (in our experiment hue values of the individual elements), these first get passed through a sigmoidal transfer function. Such a transfer leads to a compression of differences at the extremes (red and blue items) and augments differences at the category boundary (purple items).”

“**Figure S1. Full behavioral and model findings.** Columns A-E depict the empirical lines and simulated bands for A) accuracy by condition, B) confidence by condition, C) confidence frequency by accuracy, D) choice reliance effects and E) confidence reliance effects. Uncertainty for the empirical data is reflected in the error bars (2.5% and 97.5% Wald confidence intervals). Uncertainty for the simulated data is reflected in the width of the bands (2.5% and 97.5% Wald confidence intervals). We collapsed across datasets as much as possible. Each row depicts the results for one model using the best-fitting parameter combinations for this respective candidate model. F) Model recovery matrix, showing the probability that data generated under a specific model is best accounted for by that same model (diagonal) versus the other models under consideration (off-diagonal). G-H) show the confidence extension of the log posterior ratio model (c-LPR) using data for the winning model (Model 4) with parameters from a single subject from Dataset 1. G) Stimuli in the experiment were sampled pseudorandomly, meaning that the average colors of the individual stimuli were kept in narrow tolerance bands around the 4 condition mean hue (x-axis). The y-axis depicts the mean decision variable resulting from robust averaging. Dashed horizontal lines reflect the confidence criteria for blue and red stimuli. The black dot reflects the stimulus example shown in Figure 2C). H) Transformation of the mean decision variable into confidence by means of the respective confidence criteria (blue and red horizontal dashed lines).”

“**Figure S4. Confidence reliance effects split by response type.** Uncertainty for the empirical data is reflected in the error bars (2.5% and 97.5% Wald confidence intervals). The models were fit to all data including error trials, using the same mixed model approach as in the main text but replacing the split by stimulus category with response type. Post-hoc contrasts replicated the results pattern reported in the main text with a significant negative slope for red responses, $z = 40.14$, $p < .001$, a positive slope for blue responses, $z = 34.13$, $p < .001$, a stronger weight of incongruent elements compared to congruent elements, both for red responses, $z = 15.31$, $p < .001$, and for blue responses, $z = 9.84$, $p < .001$, and a stronger influence of inlying (positions 3-6) versus outlying (positions 1-2 and 7-8), both for red responses, $z = 10.78$, $p < .001$, and blue responses, $z = 11.83$, $p < .001$.”

In addition, we have added confidence intervals to Figure 3B:

A)

B)

Figure 3. Color context flips the RIE effect. *A) Schematic illustration of the stimuli used for Dataset 10, which could on average be blue, purple or red (overlapping distributions). Participants were presented with these stimuli either in the Blue Context (blue vs. purple trials) or in the Red Context (red vs. purple trials). Importantly, purple stimuli were physically identical in both contexts as indicated by the example stimuli shown spread across the color spectrum. B) Regression coefficients for the 8 sorted, colored elements of the identical purple stimuli shown either in the context of red stimuli or in the context of blue stimuli, i.e. depending on the color context. The purple example stimulus from A) is shown below the x-axis, but it should be noted that the regression coefficients shown are based on all purple stimuli from the experiment. Uncertainty is reflected in the error bars (2.5% and 97.5% Wald confidence intervals).*

We have furthermore added an explanatory sentence regarding the error bars into the Result section on p. 21: “The error bars in Figure 3B are noticeably larger than those in Figure 2. This is unsurprising, as the data in Figure 3B is based on only $N=32$ participants, whereas the previous analysis in Figure 2 was conducted on a larger sample of $N=176$ participants.”

Please note our explanation E_3 for why the sign of some of the regression curves has changed (i.e., a purely cosmetic reason as our analysis is based on the absolute weights).

R1_8) “It is a bit hard to grasp what the 'context' of the pre-registered experiment is. Qs”

Whilst we defined the colour contexts as our key conditions in the Introduction and later in the Results section, we did not consistently refer to them in capital first letters throughout the manuscript and supplement. We have changed this now to make clear that these contexts are conditions which are being contrasted:

Main manuscript, p. 7: “Participants faced identical purple stimuli within two distinct color contexts: distinguishing between blue and purple (henceforth rereferred to as the ‘Blue Context’), or purple and red (‘Red Context’). Consistent with our hypothesis, we found that the direction of the RIE effect could be reversed by the color context (i.e., for physically identical purple stimuli, confidence was driven most by bluish elements in the Blue Context and reddish elements in the Red Context).”

Main manuscript, pp. 20-21: “We therefore designed a preregistered experiment in which participants categorized the exact same purple stimuli (consisting of eight colored elements) twice: once in the context of blue stimuli composed of eight elements (i.e. Blue Context) and once in the context of red stimuli composed of eight elements (i.e. Red Context; Figure 3A). Contrasting the exact same purple elements dependent on the color context they appear in allows us to see whether physically identical elements will be assigned different confidence weights once the task-relevant feature space is shifted. In other words, do participants really base their confidence more on incongruent elements, regardless of where the elements fall in the color space? The key results of this experiment are presented in Figure 3B: First, we replicated the RIE effect in this new dataset. Second, as hypothesized, the direction of the RIE effect depended on the color context. In the Blue Context, confidence for the purple stimuli over-relied on the

‘blueish’ items. In the **Red Context**, on the other hand, confidence for the purple stimuli over-relied on the ‘reddish’ items. Post-hoc contrasts indeed found a significant negative slope for purple trials in the Blue Context, $z = 3.18, p = .001$, and a positive slope for purple trials in the **Red Context**, $z = 3.65, p < .001$.”

Main manuscript, pp. 29-30: “We manipulated the category bound using a within-subject design: participants completed 5 experimental blocks in each color context, either deciding whether stimuli were blue versus purple (**Blue Context**) or purple versus red (**Red Context**). The purple stimuli used in both contexts were identical, but we expected to find that purple in the **Blue Context** would be treated like red (i.e., blueish elements contribute more to choices and confidence) and that purple in the **Red Context** would be treated like blue (i.e., reddish elements contribute more to choices and confidence). Stimulus categories were equally frequent within each context. Each decision was followed by a confidence judgement, again using a 6-point scale. Each experimental block was 84 trials long, 24 of which were trials without confidence judgements and instead auditory error feedback allowing participants to avoid developing a strong color bias. These feedback trials were excluded from all analyses. This means we included 300 trials per color context. The order of color contexts, as well as the color key mapping, and the confidence key mapping were counterbalanced.

We had planned to create red and blue stimuli as equidistant to the purple ones, therefore creating color contexts of comparable difficulty. However, due to a coding mistake, the red stimuli were closer to the purple than the blue stimuli, creating a more difficult **Red Context**.”

Supplement, p. 9: “Despite planning to create color contexts that were matched for performance and confidence, the **Blue Context and the Red Context** significantly differed in all of these dependent variables, with the red condition being more difficult, reflected in higher error rates, $t(31) = 7.88, p < 0.001, BF = 1848168$, slower RTs, $t(31) = 5.51, p < 0.001, BF = 3947$, and lower confidence, $t(31) = 6.83, p < 0.001, BF = 7.25e+20$.”

Supplement, p. 11: “Only the purple trials in the **Red Context** were significant, $z = 2.54, p = .01$ (purple in **Blue Context**: $z = 1.42, p = .16$; blue: $z = -1.56, p = .12$; red: $z = -1.28, p = .20$). See also the **left** panels of Figure S7.”

Supplement, p. 12: “We expected to find an effect of congruency (Hypothesis 3a), which was indeed the case for the purple conditions (purple in the **Blue Context**: $z = 3.18, p = .001$; purple in the **Red Context**: $z = 3.65, p < .001$), but neither for the red, $z = -0.23, p = .82$, nor blue trials, $z = 0.49, p = .62$. We again calculated the equivalent linear contrast that we did not preregister, which showed a similar pattern with only the purple conditions showing the expected pattern (purple in the **Blue Context**: $z = 3.91, p < .001$; purple in the **Red Context**: $z = 3.40, p < .001$), but neither the red, $z = -0.05, p = .96$, nor blue trials, $z = 0.95, p = .34$. For the inlying-vs-outlying contrast (Hypothesis 3b), **only** the red **and blue** conditions showed the expected pattern (**red**: $z = 3.25, p = .001$; **blue**: $z = 2.12, p = .03$), but **not the other two**, $|zs| < 0.80, ps > .32$.”

Reviewer 2

R2_1) “Pooling data together, either for subjects or across experiments, hides potentially interesting patterns. While the authors aim to examine an overall effect of whether disconfirmatory evidence drives confidence more than confirmatory evidence (the response-incongruent effect), potential individual differences would be interesting to explore and could advance the understanding of this effect. The same applies to across-study differences, though I imagine there isn’t much heterogeneity across studies since the tasks used in all 10 experiments are essentially the same.”

We agree with the reviewer that it would be useful for the reader have some information about across-study variability. This has also been raised by Reviewer 1. Please refer to R1_3, where we have addressed this comment by including a new supplementary figure.

R2_2) “Color perception varies among individuals, affecting their category boundary for differentiating blue vs. red. This factor is related to my major comment, and I wonder how/whether it is incorporated into the results.”

The reviewer raises a good point that color perception may vary among individuals, thus affecting the category boundary. However, to prevent this from happening as much as possible in most of the experiments reported here, we used a design that strategically interspersed experimental blocks with brief feedback blocks to prevent participants from developing a strong color bias. For example, in Dataset 1 (Boldt, de Gardelle & Yeung, 2017), each experimental block was preceded by 16 trials with auditory feedback but without confidence judgments. Similarly, in our latest preregistered experiment (Dataset 10), each experimental block was preceded by 24 feedback trials without confidence judgments. In the challenging color task used here—characterized by an arbitrary category boundary that participants had to learn during practice blocks—such intermittent feedback was shown to reduce noise in the data and improve reliability. To confirm that the feedback blocks indeed helped to reduce potential biases, we relied on Signal Detection Theory to compute the decision criterion separately for each participant:

$$c = -.5 * \left(\frac{z(p(\text{hit}))}{z(p(\text{false alarm}))} \right).$$

As can be seen in the figure below (a histogram showing the number of participants with a value of c within a small range), this effectively reduced these biases, with most participants showing c values around zero.

R2_3) “In Figure 1C, the y-axis label “Average Hue” has an arrow pointing up, and Figure 1B also has an arrow next to the label “Hue.” More explanation is needed for Figure 1C.”

We have carefully considered what might have been unclear with respect to defining hue in Figure 1B and average hue in Figure 1C, however we are unsure where the confusion lies. If the reviewer could provide us with a bit more detail about the source of their confusion, we are of course happy to make this clearer.

R2_4) “Were the subjects incentivized in any way to accurately use their confidence rating?”

No, participants were never incentivised for their confidence ratings.

R2_5) “Is there a change of strategy over time? Analyzing the data split into halves might reveal if reliance on choice-incongruent evidence changes throughout the experiment, as training could affect confidence computation. For instance, Buonocore and Riccaboni (2015) found that metacognitive training can reduce the bias against dis-confirmatory evidence in patients with schizophrenia.”

We thank the reviewer for suggesting this interesting analysis. We have re-run the regression analyses for both empirical choices and confidence, splitting the data into halves of the respective experiments. As shown below, there was no clear evidence of a strategy change over time. The respective slopes remained significant in all cases, as confirmed by post-hoc analyses: for choices, $z_s > 14.2$, $p_s < 0.001$, and for confidence, $z_s > 27.0$, $p_s < 0.001$. While we have not included this additional analysis in the manuscript, we are happy to do so if Reviewer 2 recommends it.

R2_6) “Discuss how the absence of feedback in their experimental design might affect their results would potentially be interesting. Feedback has been shown to reduce bias (Haddara & Rahnev, 2022).”

The reviewer correctly notes that the data analysed in the present study comes from trials where confidence judgments were collected instead of performance feedback. However, many datasets in this study employed a design that strategically interspersed experimental blocks with brief feedback blocks to prevent participants from developing a strong colour bias. For example, in Dataset 1 (Boldt, de Gardelle & Yeung, 2017), each experimental block was preceded by 16 trials with auditory feedback but without confidence judgments. Similarly, in our latest preregistered experiment (Dataset 10), each experimental block was preceded by 24 feedback trials without confidence judgments. In the challenging colour task used here—characterised by an arbitrary category boundary that participants had to learn during practice blocks—such intermittent feedback was shown to reduce noise in the data and improve reliability.

R2_7) “In the sanity check where conditions differ in confidence but not accuracy, can the authors provide the statistical comparison for confidence? It is currently missing from the Results.”

The statistical test is already reported albeit five lines above the condition means on p. 9:

“ $\chi^2(1) = 3.69, p = .055, BF = .35$ ”

R2_8) *“Mentioning the six model variants earlier would clarify the context when Models 2 and 3 appear in Figure 2. It currently seems abrupt without prior introduction.”*

Thank you for making us aware of this issue. We now also mention the other models on pp. 9-10 where Figure 2 is introduced for the first time:

“The lines in Figure 2A – here superimposed onto simulations generated from two out of seven models further described below – show the resulting parameters from two logistic mixed models fitted to all pooled data predicting choices based on the hue values of the eight colored elements of each stimulus, separately for each color category (i.e., red or blue).”

R2_9) *“The trend reversal of regression coefficients in positions 3 and 6 for purple (vs. red) and in position 4 for purple (vs. blue) should be discussed.”*

We have added error bars to the figures displaying the results for Dataset 10, in response to Reviewer 1’s request (see R1_7). As expected, these error bars are quite large due to the small sample size ($N=32$), suggesting that the observed trend reversal is likely attributable to noise. Now that the error bars are added, we believe this point becomes much clearer, but please let us know whether you still want us to add a sentence to the manuscript.

R2_10) *“Additional thoughts on the models of confidence and decisions based on the same evidence could enhance the discussion. There are strong post-decisional models of confidence (e.g., Pleskac & Busemeyer, 2010; Moran et al., 2015; Herregods et al., 2023) and models suggesting confidence signals during initial evidence accumulation (e.g., Dotan et al., 2018; Hellmann et al., 2022; Rahnev et al., 2016; Ratcliff & Starns, 2009; Vickers, 1979; Yu et al., 2015). While the authors cite Shekhar & Rahnev (2022), expanding on these models could be beneficial.”*

Thanks for this suggestion. We have now added the following paragraph to the discussion on p. 27:

“As a related question, an increasing number of studies uses computational models to unravel how the computation of confidence unfolds over time. A key debate in this domain is whether confidence is modelled as arising only after a choice has been made (e.g., Pleskac & Busemeyer, 2010; Moran et al., 2015; Herregods et al., 2023; Yu et al., 2015) or whether confidence can already be traced back to the initial stage of evidence accumulation informing the choice (e.g., Dotan et al., 2018; Hellmann et al., 2023; Rahnev et al., 2016; Ratcliff & Starns, 2009; Vickers,

1979). Although we did not implement a dynamic model in the current work, at first glance, our findings appear to be in line with accounts positing that confidence can be traced back to the initial evidence accumulation stage. The observation that both choices and confidence depend on the same compressed evidence suggests that they are based on the same decision variable (i.e., representing the same evidence). Proponents of post-decision models, however, might argue that their models can still account for these data by assuming that confidence is based on a purely post-decision accumulation process which samples from the same compressed evidence that the choices were based on. Ultimately, implementing and comparing the above models would be needed to answer this question.”

R2_11) “The paper Shekhar & Rahnev (2022) is now published at Journal of Experimental Psychology: General. DOI number: 10.1037/xge0001524.”

Thank you for alerting us to this change. We have updated the reference accordingly on p. 40:

Shekhar, M., & Rahnev, D. (2024). How do humans give confidence? A comprehensive comparison of process models of perceptual metacognition. *Journal of Experimental Psychology: General*, 153(3), 656–688. <https://doi.org/10.1037/xge0001524>

Reviewer 1

R1_1) “Reviewer 2 and I had the same request (R1_3 and R2_1): to see whether the model accounts for individual differences instead of data averaged across participants. I don't think that Fig. S2 addresses our point about individual differences (it shows differences across datasets, not across participants). So I assume that the model does not reproduce individual participants' behavior. Since this is a limitation but does not invalidate the results, and to avoid further delaying the manuscript's acceptance, I propose that the authors simply acknowledge this in the manuscript.”

Thank you for the clarification, and apologies for the earlier misunderstanding. You're right that the previous figure showed variability across datasets rather than across participants. We've now added a new supplementary figure (Figure S6) that illustrates model fit at the individual participant level. As this shows, the model accounts well for individual differences in behaviour. We hope this directly addresses the original concern.

Main manuscript, p. 22: “As can be seen in Figure S3A and S3B, most models do a decent job at capturing average accuracy and average confidence in each of the cells of the experiments, and they do so across experiments (Figure S5) and individual participants (Figure S6).”

Figure S6. Empirical and simulated error rates across datasets and participants. Each large marker represents the average error rate (left panels) or confidence (right panels) for a given condition, with empirical values on the x-axis and simulated values on the y-axis. Marker color corresponds to the exaggerated hue used to represent the color mean condition, and marker fill indicates the variance condition. Smaller markers in matching colors show individual participants' average values for each condition. Each row of panels corresponds to a different dataset.